# Force propagation between epithelial cells depends on active coupling and mechano-structural polarization

Artur Ruppel[1†‡], Dennis Wörthmüller[2,3†], Vladimir Misiak[1], Manasi Kelkar[4], Irène Wang[1], Philippe Moreau[1], Adrien Méry[1], Jean Révilloud[1], Guillaume Charras[4,5,6], Giovanni Cappello[1], Thomas Boudou[1], Ulrich S Schwarz[2,3*], Martial Balland[1*]

[1]Université Grenoble Alpes, CNRS, LIPhy, Grenoble, France; [2]Institute for Theoretical Physics, Heidelberg University, Heidelberg, Germany; [3]BioQuant–Center for Quantitative Biology, Heidelberg University, Heidelberg, Germany; [4]London Centre for Nanotechnology, University College London, London, United Kingdom; [5]Department of Cell and Developmental Biology, University College London, London, United Kingdom; [6]Institute for the Physics of Living Systems, University College London, London, United Kingdom

*For correspondence:
schwarz@thphys.uni-heidelberg.de (USS);
martial.balland@univ-grenoble-alpes.fr (MB)

†These authors contributed equally to this work

Present address: ‡Centre de Recherche en Biologie Cellulaire de Montpellier, Centre National de la Recherche Scientifique, Montpellier, France

Competing interest: The authors declare that no competing interests exist.

**Abstract** Cell-generated forces play a major role in coordinating the large-scale behavior of cell assemblies, in particular during development, wound healing, and cancer. Mechanical signals propagate faster than biochemical signals, but can have similar effects, especially in epithelial tissues with strong cell–cell adhesion. However, a quantitative description of the transmission chain from force generation in a sender cell, force propagation across cell–cell boundaries, and the concomitant response of receiver cells is missing. For a quantitative analysis of this important situation, here we propose a minimal model system of two epithelial cells on an H-pattern ('cell doublet'). After optogenetically activating RhoA, a major regulator of cell contractility, in the sender cell, we measure the mechanical response of the receiver cell by traction force and monolayer stress microscopies. In general, we find that the receiver cells show an active response so that the cell doublet forms a coherent unit. However, force propagation and response of the receiver cell also strongly depend on the mechano-structural polarization in the cell assembly, which is controlled by cell–matrix adhesion to the adhesive micropattern. We find that the response of the receiver cell is stronger when the mechano-structural polarization axis is oriented perpendicular to the direction of force propagation, reminiscent of the Poisson effect in passive materials. We finally show that the same effects are at work in small tissues. Our work demonstrates that cellular organization and active mechanical response of a tissue are key to maintain signal strength and lead to the emergence of elasticity, which means that signals are not dissipated like in a viscous system, but can propagate over large distances.

## Editor's evaluation

Using surface micropatterning, optical activation, and theoretical analysis, the authors provide compelling evidence that adjacent cells actively propagate mechanical stress in epithelial tissues. The response of the receiver cell is active and enhanced when the principal stress direction is perpendicular to the orientation of actin fibers. This work is important and a must read for everybody wanting to understand tissue mechanics.

## Introduction

Cell-generated forces are essential for tissue morphodynamics, when eukaryotic cells change number, shape, and positions to build a multicellular tissue. Tissue morphogenesis is a dominant process during development, but also occurs in adult physiology and disease, in particular during wound healing and cancer, respectively. In addition to driving cell shape change and movement, force-producing processes allow cells to probe the mechanical and geometrical properties of their environment (*Discher et al., 2005*; *Luciano et al., 2021*), feeding back on to major cellular processes, such as differentiation (*Engler et al., 2006*; *McBeath et al., 2004*; *Wen et al., 2014*; *Kilian et al., 2010*), fate (*Chen et al., 1997*; *Théry et al., 2005*; *Nelson et al., 2005*), or migration (*Pathak and Kumar, 2012*; *Sunyer and Trepat, 2020*; *Shellard and Mayor, 2021*). Generation of contractile force is a universal property of mammalian cells due to the ubiquitous expression of non-muscle myosin II (*Vicente-Manzanares et al., 2009*). It is less clear, however, how this force is propagated through tissues and how long-ranged its effects are. Fast and long-ranged propagation of mechanical force seems to be essential during development, when morphogenesis has to be coordinated across the embryo (*Ho et al., 2019*; *Desprat et al., 2008*). For example, the onset of migration of neural crest cells in *Xenopus* appears controlled by the stiffening of the underlying mesoderm resulting from axis elongation (*Barriga et al., 2018*). An example of a more mature tissue is the epithelium of the juvenile esophagus in mice, whose transition from growth to homeostasis is mediated by the mechanotransduction of progressively increasing mechanical strain at the organ level (*McGinn et al., 2021*).

Despite these interesting observations for development, it is not clear how force is propagated across tissues in general and whether propagation is passive or sustained by mechanochemical feedback loops. Force propagation across tissues suffers from the same challenge as any other information propagation through a passive medium. Whether it be an electrical signal transmitted through a telegraph line or an action potential originating in the soma of a neuron, the signal typically attenuates with distance until it becomes indistinguishable from noise (*Kholodenko et al., 2010*). The main measure to counteract such attenuation are active processes that restore signal strength, like the opening of voltage-gated ion channels along the axon for action potentials. In addition to electrical currents, mechanical waves have also been observed to propagate along lengths several orders of magnitude larger than the cell size in confined epithelial tissues (*Di Talia and Vergassola, 2022*). These waves require active cellular behaviors such as contractility and F-actin polymerization to propagate, suggesting that cells actively respond to external forces to maintain the strength of the signal as it propagates through the tissue (*Serra-Picamal et al., 2012*; *Peyret et al., 2019*; *Petrolli et al., 2019*). Furthermore, it has been shown that passive cells in an epithelial tissue act as obstacle for mechanical wave propagation (*Ng et al., 2014*). Despite these studies, our knowledge of force propagation remains largely qualitative because of the lack of a model system that allows for precise spatiotemporal control of force generation and quantitative characterization of the propagation of the mechanical signal across intercellular junctions. As a result, we know little of how far force signals can propagate from their origin or whether signal propagation efficiency depends on tissue organization. Indeed, in some tissues, such as the hydra ectoderm, stress fibers within the cells of the ectoderm form a nematic system (*Maroudas-Sacks et al., 2021*). This high degree of alignment of force-generating subcellular structures suggests that tissues may display anisotropic propagation of stresses.

Here, we introduce such a sought-after minimal biophysical system for force propagation in epithelia, consisting of two interacting cells in which force generation is controlled by an optogenetic actuator of contractility and force propagation is quantitatively monitored using traction and monolayer force microscopies. To place the two cells next to each other with a stable cell–cell boundary, we make use of adhesive micropatterning (*Théry et al., 2006*; *Mandal et al., 2014*). Moreover, the adhesive micropatterning allows us to control the aspect ratio of the cells and the structural organization of their cytoskeleton. Using this system, we show that intercellular force propagation is an active mechanism, with the receiver cell actively adapting to the signal from the sender cell. We then demonstrate how the degree of active coupling is controlled by key morphological parameters, such as junction length and the degree and orientation of mechanical polarization. Strikingly, force propagation is amplified perpendicularly to the axis of mechano-structural polarization, similar to the Poisson effect in passive material. Finally, we verify that our findings in these cell doublets can be generalized to larger cell clusters. Overall, we show that active cellular responses to incoming forces can maintain signal strength and lead to the emergence of an apparent elastic behavior that allows

signals to be propagated over large distances, as in an elastic material, rather than be dissipated, as in a viscous material.

## Results

### The intercellular junction decreases the mechano-structural polarization

The most important feature of epithelial tissue is strong cell–cell adhesion, which makes the epithelial monolayer a coherent sheet that can effectively separate different compartments, like the outside and inside of a body or organ. Therefore, we first characterized how the presence of an intercellular junction influences cellular organization and force generation. To this end, we compared cell pairs ('doublets') with single cells ('singlets') grown on identical micropatterns (*Figure 1A*). The H-pattern is known to be able to accommodate both doublets and singlets, which in both cases form an hour-glass shape (*Figure 1B and C*, respectively). Note that most doublets form from a single cell that has divided on the pattern.

We found that when plated on H-shaped micropatterns, singlets formed prominent stress fibers around the cell contour (peripheral stress fibers), as well as some smaller internal stress fibers which resulted from the spreading process (*Figure 1C* and *Figure 1—figure supplement 1*). Vertical stress fibers at the edge of the patterns, along the vertical bars of the H, were straight and strongly coupled to the substrate (adherent stress fibers), while peripheral stress fibers located above the non-adhesive regions of the micropattern, in between the vertical bars of the H, were curved due to the inward pull of the cell cortex (free stress fibers). Focal adhesions were primarily located in the corners of the pattern, although some were present on the middle bar of the H-pattern, which is required for the cells to spread over the whole pattern. A similar pattern of organization was observed in doublets, with the addition of a prominent cell–cell junction in the center of the H-pattern, parallel to the lateral bars of the H (*Figure 1B*), consistent with previous work (*Tseng et al., 2012*).

By quantifying cell-generated forces using traction force microscopy (TFM), we found that the magnitude of traction forces is very similar between doublets and singlets (*Figure 1D*). When we quantified the overall contractility by calculating the strain energy stored in the substrate, we found that it is even slightly higher for singlets than for doublets, despite spreading over the same surface area (*Figure 1F*). This is likely because singlets have to spread a smaller volume over the same surface as doublets, leading to higher tension, both in the actomyosin machinery (*Hippler et al., 2020*) and in the cell membrane (*Pontes et al., 2017*). Moreover, they do not have to accommodate any cell–cell junction and therefore could be coupled better to the substrate (*Tseng et al., 2012*).

Next we calculated stresses born by the cells using monolayer stress microscopy (MSM), which converts the TFM data into an estimate for intracellular stress (*Figure 1E*, *Tambe et al., 2011*; *Bauer et al., 2021*). Although MSM assumes linear elasticity of the cell layer (*Tambe et al., 2013*), it is generally believed to give a good representation of the spatial distribution of stress in the interior of adherent cells (*Ng et al., 2014*). In doublets, the normal stresses in x- and y-direction ($\sigma_{xx}$ and $\sigma_{yy}$) were comparable, whereas in singlets $\sigma_{xx}$ was much larger than $\sigma_{yy}$ (*Figure 1F*). To quantitatively compare the cellular stress distribution of these systems, we computed the mechanical polarization as $(\sigma_{xx} - \sigma_{yy})/(\sigma_{xx} + \sigma_{yy})$. With this quantification, a system polarized vertically has a polarization of −1, 0 reflects an unpolarized system and 1 a horizontally polarized system. Doublets were unpolarized (average degree of polarization of 0), whereas singlets were horizontally polarized with an average degree of polarization of almost 0.5. Next, we measured the polarization of the actin structures with a homemade algorithm using the structure tensor (see 'Materials and methods' section for details). We found the same trend and a strong correlation between mechanical and structural polarization, meaning that the stress fibers in singlets are largely organized horizontally, whereas in doublets they are directed more toward the center (*Figure 3—figure supplement 1*). Our results suggest that intercellular junctions may act as a barrier preventing the horizontal organization of stress fibers that exist in singlets, thus strongly altering the mechanical polarization of the system.

### The presence of an intercellular junction leads to a redistribution of tension from free to adherent peripheral stress fiber

An inherent limitation of TFM is that it only quantifies tension transmitted to the substrate while forces internally balanced are not detected. Although this is partially remedied by MSM, which estimates an

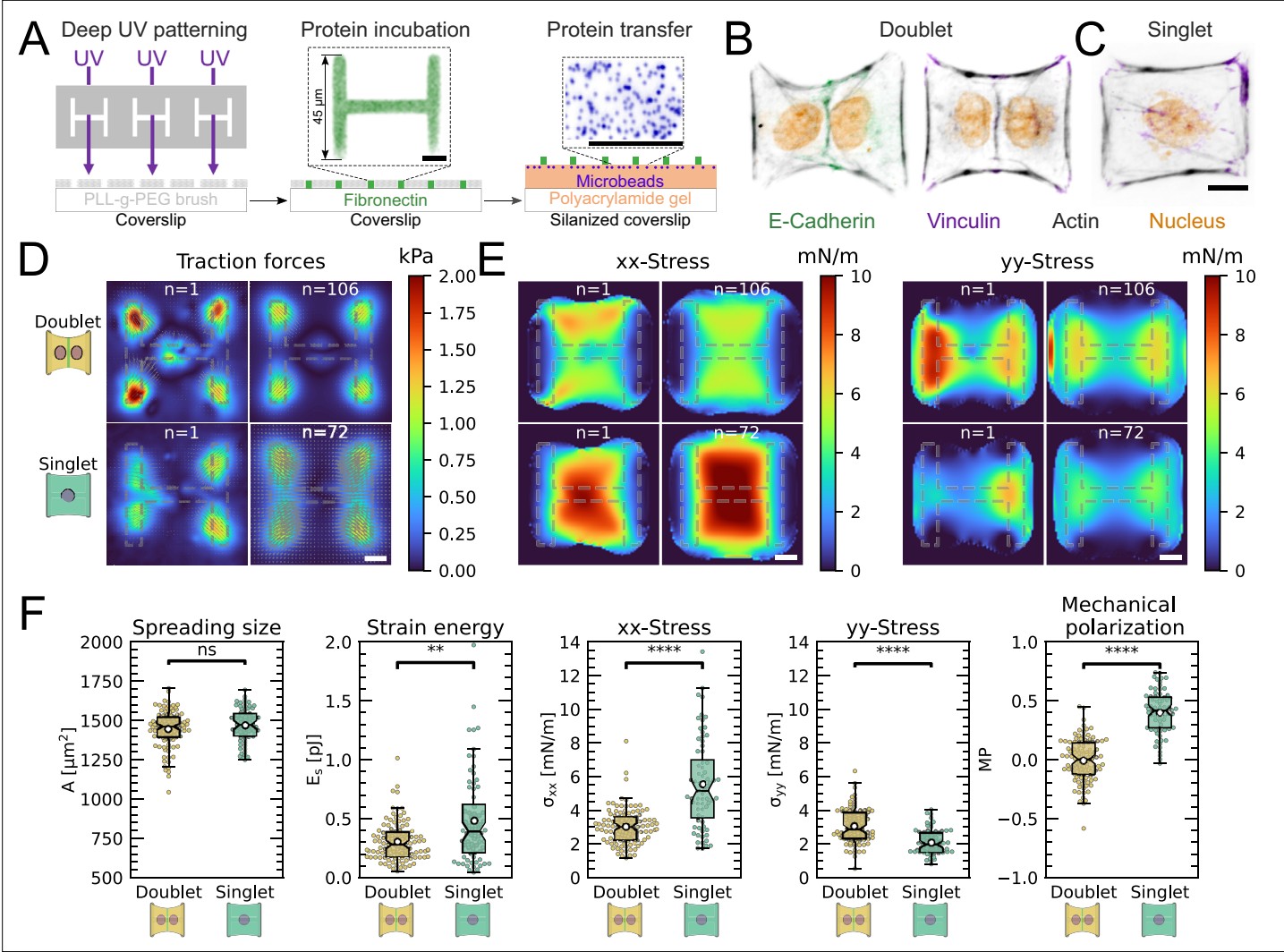

**Figure 1.** The cell–cell junction leads to a decrease in mechanical polarization. (**A**) Cartoon of the micropatterning process on soft substrates, allowing to control cell shape and measure forces at the same time by embedding fluorescent microbeads into the gel and measuring their displacement. The middle panel shows the used pattern geometry, an H with dimensions of $45\,\mu m \times 45\,\mu m$. (**B**, **C**) Immunostaining of opto-MDCK cells plated on H-patterns and incubated for $24h$ before fixing. Actin is shown in black, E-cadherin in green, vinculin in violet, and the nucleus in orange. (**B**) The left and right images show a representative example of a doublet. (**C**) A representative example of a singlet. (**D**) Traction stress and force maps of doublets (top) and singlets (bottom) with a representative example on the left and an average on the right. (**E**) Cell stress maps calculated by applying a monolayer stress microscopy algorithm to the traction stress maps, with a representative example on the left and an average on the right. (**F**) From left to right, boxplots of spreading size, measured within the boundary defined by the stress fibers. Strain energy, calculated by summing up the squared scalar product of traction force and displacement field divided by two xx-stress and yy-stress calculated by averaging the stress maps obtained with monolayer stress microscopy. Degree of polarization, defined as the difference of the average xx- and yy-stress normalized by their sum. Doublets are shown in yellow and singlets are shown in green. The figure shows data from n = 106 doublets from N = 10 samples and n = 72 singlets from N = 12 samples. All scale bars are $10\,\mu m$ long.

The online version of this article includes the following figure supplement(s) for figure 1:

**Figure supplement 1.** Immunostaining of opto-MDCK cells plated on H-patterns and incubated for $24hr$ before fixing.

internal stress distribution based on the TFM results, this method lacks spatial resolution to take into account the precise organization of the cell. In order to address this important aspect, we therefore turn to a contour model (CA) that focuses on the role of the peripheral stress fibers (*Figure 2A*).

We previously showed that the curvature of a free stress fiber results from a balance between an isotropic surface tension pulling the stress fibers toward the cell center and a line tension acting along the fibers, tending to straighten them (*Bischofs et al., 2008*; *Bischofs et al., 2009*). The radius of curvature is then given by the ratio of the line to the surface tension. As the line tension can be

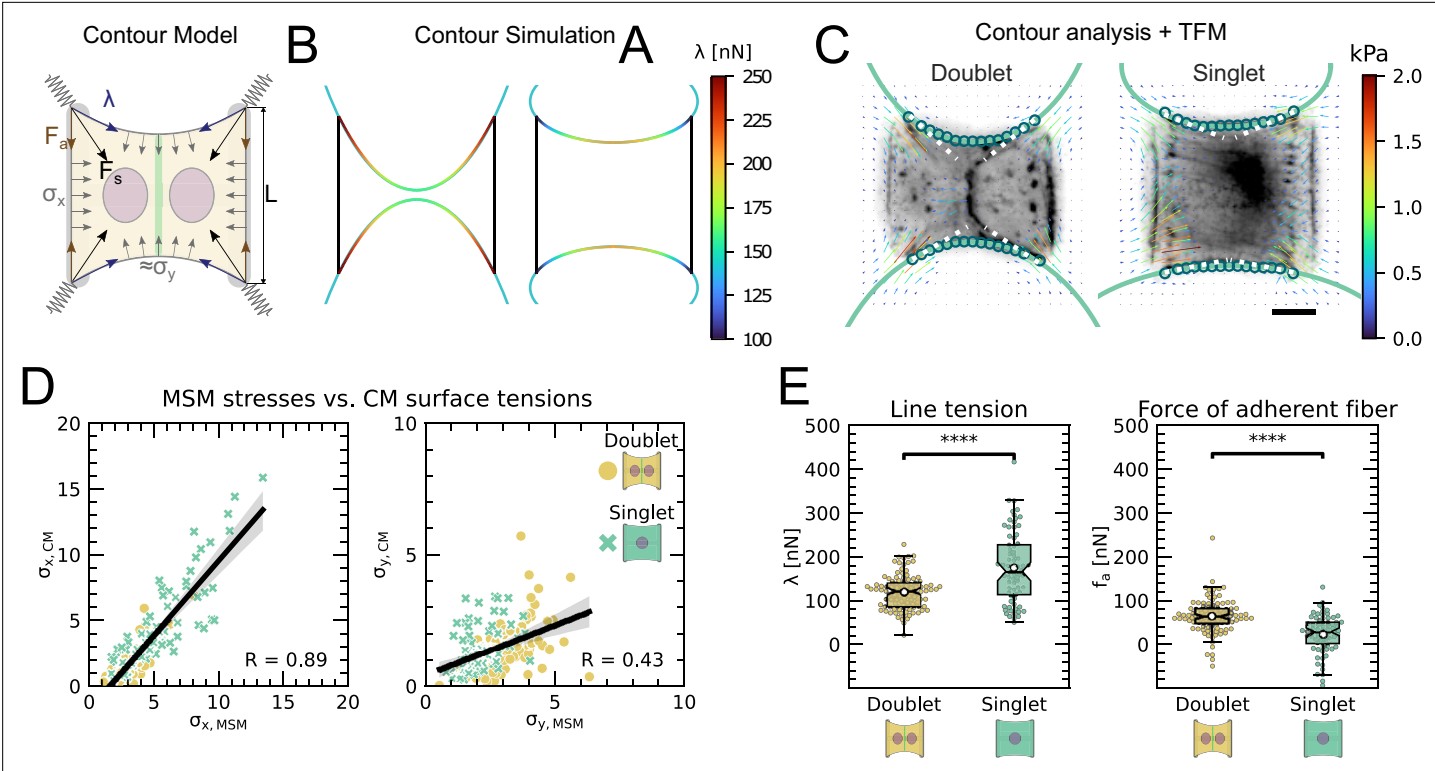

**Figure 2.** The cell–cell junction leads to a redistribution of tension from free to adherent peripheral stress fiber. (**A**) Cartoon of the contour model used to analyze the shape of the doublets and singlets. (**B**) Finite element method (FEM) simulation of the contour with $\sigma_y > \sigma_x$ left and $\sigma_x > \sigma_y$ right. (**C**) Actin images of doublets (left) and singlets (right) with traction stresses (arrows), tracking of the free fiber (blue circles), elliptical contour fitted to the fiber tracks (green line), and tangents to the contour at adhesion point (white dashed line). The scale bar is $10\,\mu m$ long. (**D**) Correlation plot of monolayer stress microscopy (MSM) stresses and CM surface tensions. MSM stresses were calculated by averaging the stress maps obtained with monolayer stress microscopy, and the surface tensions were obtained by the contour model analysis, where $\sigma_x$ was measured on the traction force microscopy (TFM) maps by summing up the x-traction stresses in a window around the center of the vertical fiber and $\sigma_y$ was determined by fitting the resulting ellipse to the tracking data of the free fiber. Doublets are shown as yellow dots, and singlets are shown as green crosses. The black line shows the linear regression of the data, and the shaded area shows the 95% confidence interval for this regression. The R-value shown corresponds to the Pearson correlation coefficient. (**E**) Boxplots of line tension $\lambda$ (left) and force of adherent fiber $F_a$ (right) as defined in panel (**A**). Both values were calculated by first calculating the force in each corner by summing up all forces in a radius of $12\,\mu m$ around the peak value and then projecting the resulting force onto the tangent of the contour for the line tension and onto the y-axis for the force of adherent fiber. Doublets are shown in yellow, and singlets are shown in green. The figure shows data from n = 106 doublets from N = 10 samples and n = 72 singlets from N = 12 samples. All scale bars are $10\,\mu m$ long.

calculated from the TFM data and the radius of curvature can be measured, the surface tension can be inferred. One key assumption of our previous work was that cellular tension is isotropic. As we showed that single cells are mechanically polarized (*Figure 1F*), we generalized our circular arc model to anisotropic systems (anisotropic tension model [ATM]) (*Pomp et al., 2018*), allowing to compute surface tensions in the x- and y-directions by measuring the surface tension in x-direction on the TFM maps and then fitting the surface tension in y-direction until the resulting ellipse fits to the fiber (*Figure 2A* and theory supplement).In the ATM, line tension $\lambda$ becomes position-dependent, as seen in *Figure 2B*; on the left-hand side, a large value of $\sigma_y$ pulls the contour in, while on the right-hand side, a smaller value of $\sigma_y$ leads to less invaginated cell contour. In both cases, one clearly sees that the contour is not circular, but elliptical (*Pomp et al., 2018*). The color code shows that anisotropic surface tension comes with spatial variation in the line tension. Application of this approach to experimental data allowed us to infer anisotropic surface tensions for both doublets and singlets (*Figure 2C*).

The combination of contour analysis and TFM showed that stress fibers in singlets are subjected to a larger stress along the x-direction than in doublets and conversely that stress fibers in doublets are subjected to higher stresses in the y-direction than singlets (*Figure 2D*). Consistent with this, singlets possessed a significantly larger line tension in their free stress fibers than doublets. In contrast, the force exerted by adherent stress fibers displayed the opposite behavior: it was higher in doublets

than in singlets (*Figure 2E*). These two forces were computed by integrating the traction stresses in each corner, correcting for the contribution of the surface tension along the adherent fiber and then projecting these forces onto the stress fibers (see theory supplement for details). These results are consistent with the MSM analysis from *Figure 1*. It should be noted that the anisotropic surface tensions obtained from the contour model are not directly related to MSM measurements since MSM focuses on the bulk and the contour models on the boundaries. Yet, a strong correlation between MSM measurements and anisotropic surface tensions was found (*Figure 2D*), suggesting that there is some indirect relationship between the two. Indeed, $\sigma_{xx}$ (which corresponds roughly to the free stress fiber since it is approximately parallel to the x-axis) is higher in singlets and $\sigma_{yy}$ (which corresponds roughly to the adherent stress fiber since it is parallel to the y-axis) is higher in doublets (*Figure 1E*). We conclude that the presence of cell–cell junction leads to a redistribution of tension from the free to adherent peripheral stress fibers.

## Force increase through local activation of RhoA in one cell leads to active force increase in neighboring cell in doublets

In order to study signal propagation, it is important to generate a well-defined input whose propagation can be followed in space and time. Although this is a notoriously difficult issue in cellular force generation, a new tool was recently established which allows just that, namely non-neuronal optogenetics. In order to switch on cell contractility in a controlled manner, we activated RhoA, a major regulator of cell contractility, with an optogenetic actuator that relocalizes a RhoGEF domain to the membrane in response to $488\,\mathrm{nm}$ light (*Figure 3A*, *Valon et al., 2017*; *Andersen et al., 2023*; *Méry et al., 2023*).

As previous work has shown that this tool allows localized activation of RhoA signaling within single cells (*Valon et al., 2015*) and we used it to activate the left half of doublets and singlets to determine how the localized stress created by activation propagated to the other side of the system. First, we compared global photoactivation of doublets and singlets (shown in Animations 1–3) with their local photoactivation (*Figure 3—figure supplement 2*). Then, to make sure we do not accidentally activate the right cell, we first looked at CRY2 recruitment in left and right cell after photoactivation of only the left cell for different light intensities (Animation 4). We identified $0.9\,\mathrm{mW\,mm^{-2}}$ to be the right intensity where recruitment in the left cell is saturated and recruitment in the right cell is much smaller. To further minimize photoactivation of the right cell, we estimated how much light it receives by measuring the intensity profile of the photoactivation region (*Figure 3—figure supplement 3A*) and saw that the light intensity right at the border of the activation region is still at 50% of its maximal value. We then decided to move the activation region $10\,\mu m$ away from the junction because there the intensity drops to 6%, that is, $0.054\,\mathrm{mW\,mm^{-2}}$. The light the right cell receives in this condition is less than the first activation seen in *Figure 3—figure supplement 2*, where no recruitment of the right cell was measured. Finally, to make sure that the light seen by the right cell is not sufficient to trigger a force response, we globally photoactivated a doublet with $0.054\,\mathrm{mW\,mm^{-2}}$ and then locally only the left cell with $10\,\mu m$ distance from the center, with $0.9\,\mathrm{mW\,mm^{-2}}$ (*Figure 3—figure supplement 3B*). The right cell sees more light in the first condition, but a force response was only measured in the second condition, so we concluded that stray light activation cannot explain the force increase of the nonactivated cell.

The stress propagation differed markedly between doublets and singlets. In doublets, traction forces increased both in the activated and the nonactivated region. In the singlets, on the other hand, traction forces increased slightly and very locally in the activated region, but decreased in the nonactivated region (*Figure 3C and D*, Animations 4 and 5). We conclude that in contrast to singlets, doublets can establish stable contraction patterns under half-activation of contractility. To rule out that this is simply an effect of different expression levels of optogenetic receptors and actuators in doublets vs. singlets, we compared photoactivation of the whole doublet with photoactivation of the whole singlet. Here, the relative strain energy increase was very similar between doublets and singlets (*Figure 3—figure supplement 2*).

We hypothesized that this behavior may originate from differences in the reorganization of contractile elements within the cytoskeleton in singlets and doublets. Therefore, we imaged the behavior of the actin cytoskeleton during the light stimulation by comparing the fluorescence intensity distribution of the F-actin reporter LifeAct before and during stimulation. In doublets, LifeAct fluorescence

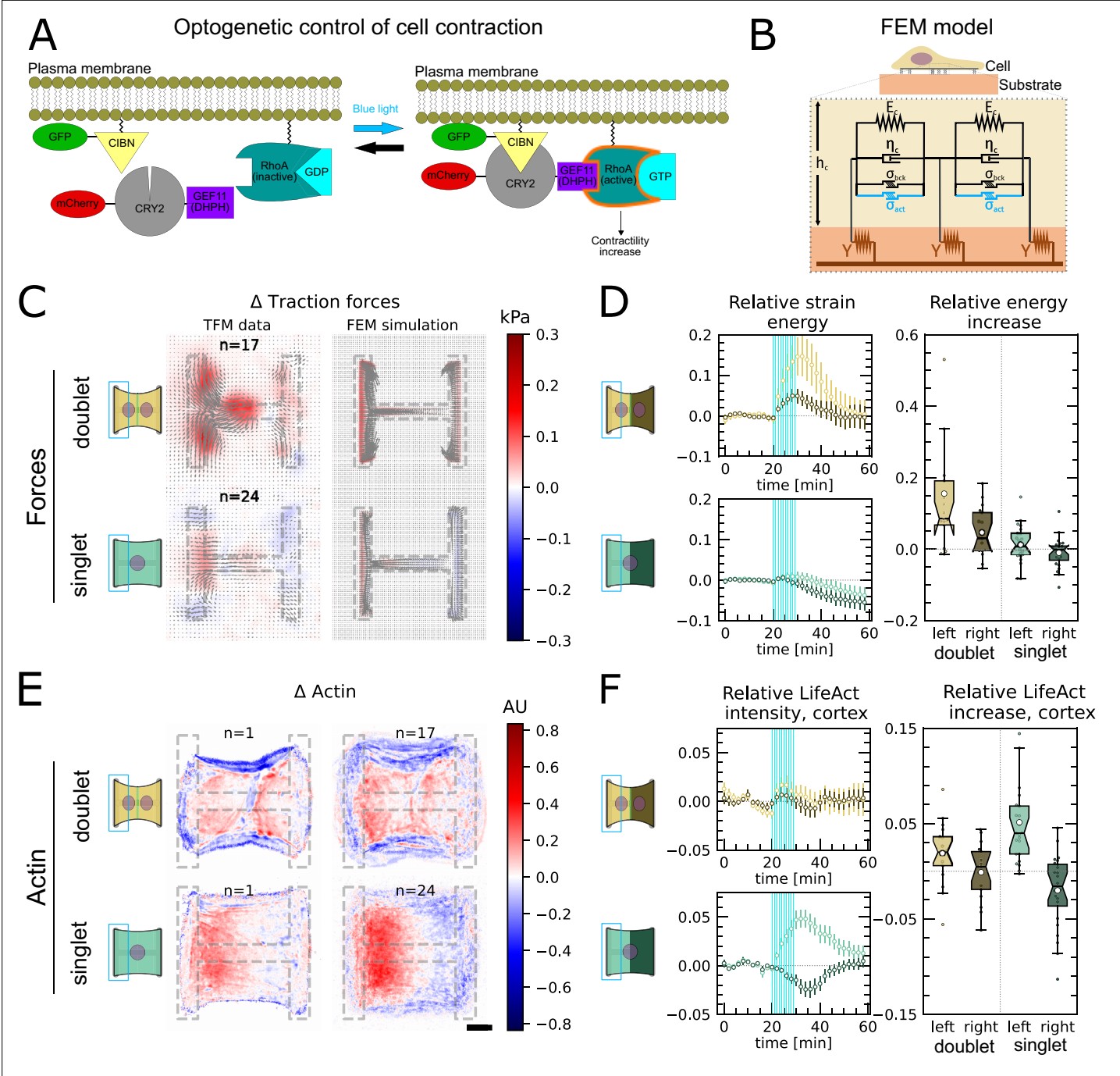

**Figure 3.** Local activation of RhoA leads to stable force increase in both the activated and the nonactivated cell in doublets, but destabilizes force homeostasis in singlets. (**A**) Cartoon of the optogenetic CIBN/CRY2 construction used to locally activate RhoA. (**B**) Cartoon of the FEM continuum model used to explain optogenetic experiments. (**C**) Difference of average traction force maps after and before photoactivation of cell doublets (top) and singlets (bottom). Maps on the left show the traction force microscopy (TFM) data, and maps on the right show the result of the FEM simulations with an active response of the right cell. (**D**) Relative strain energies of doublets (top) and singlets (bottom) with local photoactivation, divided into left half (bright) and right half (dark). One frame per minute was acquired for 60 min, and cells were photoactivated with one pulse per minute for 10 min between minute 20 and minute 30. Strain energy curves were normalized by first subtracting the individual baseline energies (average of the first 20 min) and then dividing by the average baseline energy of all cell doublets/singlets in the corresponding datasets. Data is shown as circles with the mean ± SEM. Boxplots on the right show the value of the relative strain energy curves 2 min after photoactivation, that is, at minute 32. (**E**) Difference of actin images after and before photoactivation of doublets (top) and singlets (bottom), with an example on the left and the average on the bottom. All scale bars are $10\,\mu m$ long. (**F**) LifeAct intensity measurement inside the cells over time (left) of left half (bright) vs. right half (dark) of doublets (top) and

*Figure 3 continued*

singlets (bottom) after local photoactivation. Boxplots on the right show the relative actin intensity value after 2 min after photoactivation of activated vs. nonactivated half. The figure shows data from n = 17 doublets from N = 2 samples and n = 17 singlets from N = 6 samples. All scale bars are $10\,\mu m$ long.

The online version of this article includes the following figure supplement(s) for figure 3:

**Figure supplement 1.** LifeAct images of doublets and singlets on H-patterns, quantification of mechano-structural polarization and measurement of LifeAct intensities of stress fibers in response to optogenetical activation of RhoA.

**Figure supplement 2.** Relative strain energies of local (left) vs. global (right) photoactivation of doublets (top) and singlets (bottom).

**Figure supplement 3.** Recruitment of CRY2 to the membrane in left vs. right cell in response to optogenetic activation with varying light intensities.

**Figure supplement 4.** Strain energy increase of non-activated cell is not caused by accidental photoactivation through stray light.

**Figure supplement 5.** Acute fluidization of actin structures in singlets is a plausible explanation of the observed strain energy curves in response to local optogenetic activation of RhoA.

increases slightly inside and decreases slightly outside of the doublet. The decrease outside of the doublet is mostly due to fiber movement. When we measure the LifeAct intensity following its movement, the intensity remains mostly constant (*Figure 3—figure supplement 1C*). In contrast, in singlets, LifeAct fluorescence redistributed from the unstimulated side to the stimulated side, both inside of the cell as well as on the periphery (*Figure 3E and F*, *Figure 3—figure supplement 1C*).

To determine whether the behavior of the doublets could arise from a passive response of the nonactivated region, we developed a finite element (FE) continuum model to predict stress propagation (*Figure 3B*) (details in theory supplement). Based on previous work characterizing cell rheology (*Edwards and Schwarz, 2011*; *Banerjee and Marchetti, 2012*; *Oakes et al., 2014*; *Oakes et al., 2017*), our continuum model consists of a network of Kelvin–Voigt elements that are each connected to an elastic substrate. Each Kelvin–Voigt element also possesses an active element, which describes the contractility of myosin motors that can be increased to simulate optogenetic activation of contractility. In order to fix the parameters of the model, we performed an experiment where we photoactivated the whole singlet/doublet (see theory supplement and Animations 1–3 for details). We used this model to predict the spatiotemporal evolution of traction stress in the system. Comparison of the FE results to the experimental data shows that the behavior of the nonactivated region cannot be reproduced with a purely passive reaction (*Figure 3—figure supplement 5A and B*). Therefore, we hypothesized that active coupling takes place perhaps due to mechanotransductory signaling pathways. To test this idea, we introduced an active coupling element into the FEM model between the left and the right half. We then used this coupling term as a fitting parameter to qualitatively reproduce the experimental traction maps. Again, doublets differed from singlets. Coupling in doublets was positive, meaning that the right half contracts in response to the contraction of the left half; whereas it was negative in singlets, meaning that the right half relaxes in response to the contraction of the left half.

Together, these data indicate that cells in the doublet are actively coupled, with the unstimulated cell responding to the contraction of the stimulated cell by actively contracting, in

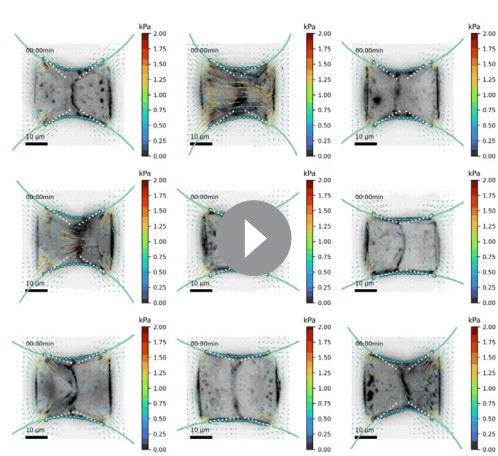

**Animation 2.** Nine examples of globally photoactivated doublets. Actin is shown in black, traction forces are overlaid as colored arrows, the tracked contour in blue circles, the tangents in white dashed lines, and the fitted ellipse in green.

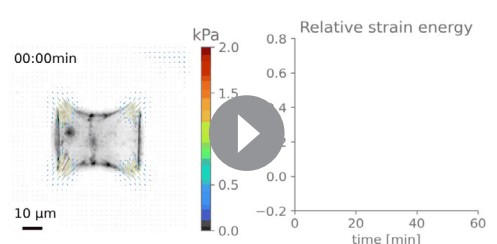

**Animation 1.** Actin + traction forces (left) and relative strain energy (right) over time of a globally photoactivated doublet.

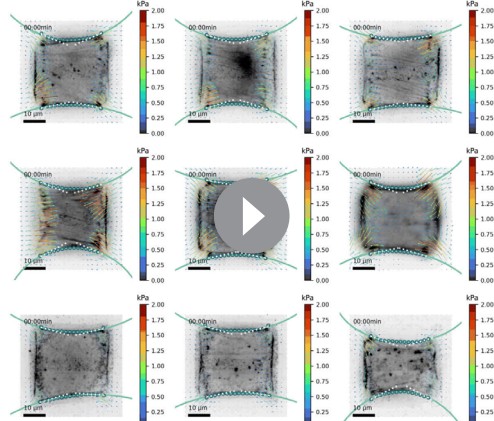

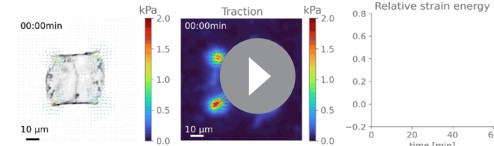

**Animation 4.** Actin + traction forces (left), traction force map (center), and relative strain energy divided in left (blue) and right (orange) half (right) over time of a locally photoactivated doublet.

**Animation 3.** Nine examples of globally photoactivated singlets. Actin is shown in black, traction forces are overlaid as colored arrows, the tracked contour in blue circles, the tangents in white dashed lines, and the fitted ellipse in green.

agreement with previous qualitative reports (*Liu et al., 2010*; *Hino et al., 2020*). Strikingly, traction force generated by doublets shows a homeostatic response to this transient increase of RhoA activity. Indeed, once activation is stopped, the traction force generated on the pattern returns to its initial level. In singlets on the other hand, transient and local RhoA activation has a destabilizing effect. The local increase in traction stress and the local accumulation of F-actin in the photoactivated region is compensated with a decrease in stress and F-actin in the nonactivated region. Furthermore, rather than displaying a homeostatic behavior, the traction stress keeps decreasing even after the activation is stopped. We hypothesize that this may occur because the actin structures acutely fluidize in response to the local stress increase, as previously reported (*Krishnan et al., 2009*; *Andreu et al., 2021*). Since there is no junction and thus no barrier for mass transport in singlets, the imbalance in stress induced by optogenetic activation may lead to a flow of F-actin from the nonactivated to the activated region, consistent with our observations (*Figure 3E and F*). As a qualitative test, we exchanged the Kelvin–Voigt elements in our model of the cell body for Maxwell elements after photoactivation. This led to a behavior consistent with our observations (*Figure 3—figure supplement 5C and D*).

Overall, our data show that the cytoskeleton possesses active coupling and that the degree of coupling depends on the presence of an intercellular junction. The intercellular junction allows efficient propagation of stress across the whole micropattern, probably due to mechanotransductory pathways and by impeding fluidization.

## Strong active coupling is present in the actin cortex of doublets

Having shown that the unstimulated cell in doublets reacts actively to the contraction of the stimulated cell, we sought to quantify the strength of this active response. To this end, we sought to quantitatively reproduce the distribution of cell stresses obtained by MSM in photoactivated doublets using our FEM model (*Figure 4A–C*). To simulate optogenetic activation, we increased the level of contractility of the activated left-hand side of the doublet compared to the baseline found in unstimulated conditions. Then to simulate coupling, we tuned the degree of contractility on the unstimulated right-hand side of the doublet. The ratio of contractility of the right half to the left half corresponds to the degree of active coupling between the cells in the doublet. An active coupling of 0 means no contraction of the right half, 1 indicates a contraction of the right half of the same magnitude as the left, and −1 means relaxation of the right half with same magnitude as the increase on the left. To allow comparison of experiments to simulations, we normalize the stress increase of the right cell by the total stress increase (*Figure 4C*). For each experiment, we determined the degree of coupling that best reproduced the experimental cellular stress distribution in the x- and y-directions (*Figure 4B*).

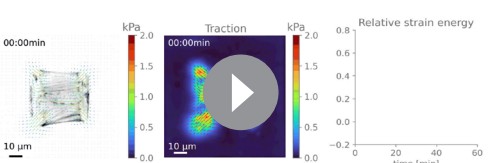

**Animation 5.** Actin + traction forces (left), traction force map (center), and relative strain energy divided in left (blue) and right (orange) half (right) over time of a locally photoactivated singlet.

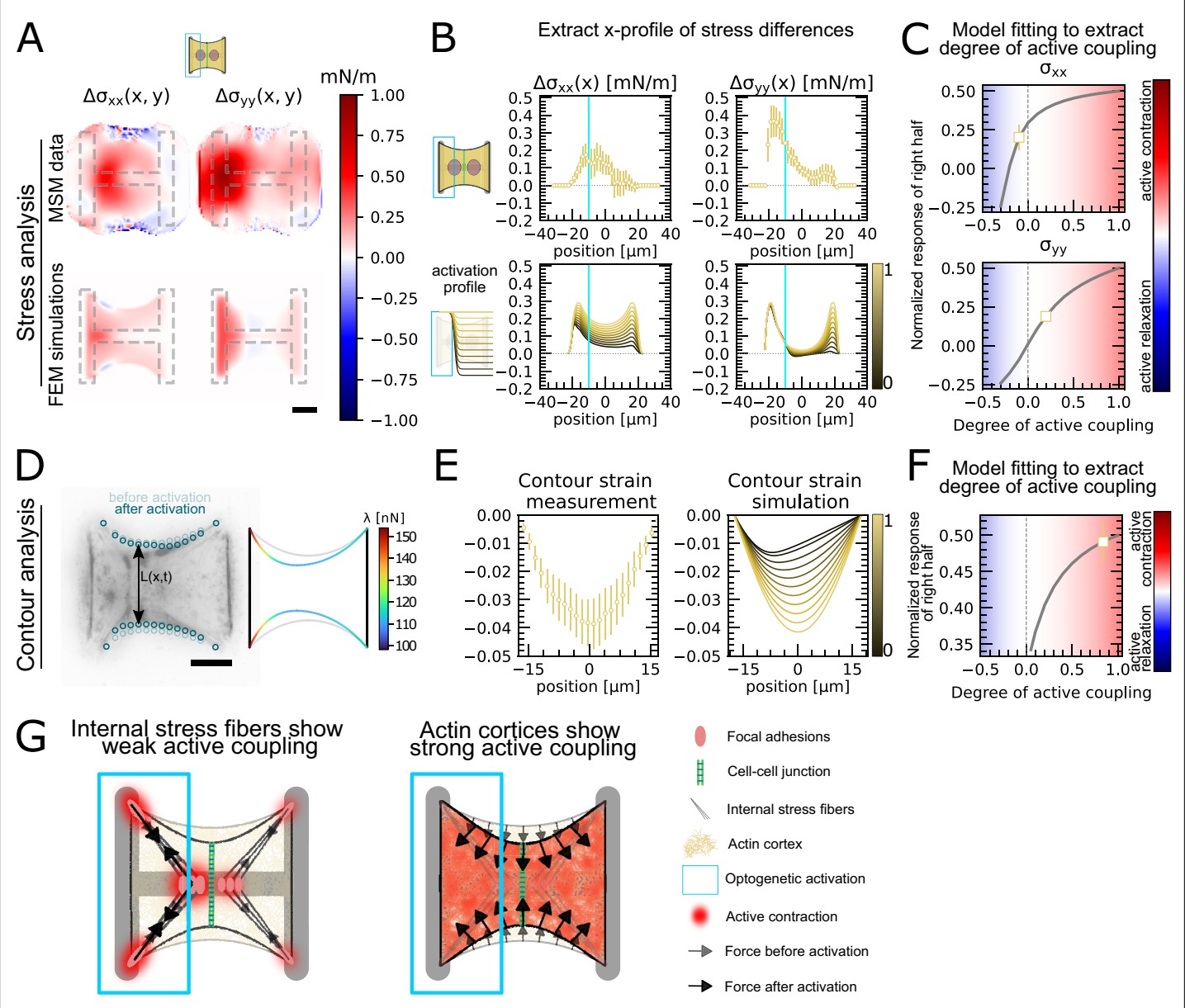

**Figure 4.** Stress and contour modeling shows strong active coupling of actin cortices in doublets. (**A**) Difference of average cell stress maps after and before photoactivation of cell doublets, calculated with monolayer stress microscopy (MSM) (top) and simulated with the FEM continuum model (bottom). Stress in x-direction is shown on the left, and stress in y-direction is shown on the right. (**B**) Average over the y-axis of the maps in (**A**). Data is shown as circles with the mean ± SEM. In the simulation, the right half of the cell was progressively activated to obtain the family of curves shown in the bottom. (**C**) Response of the right half (normalized by the total response), obtained from the model (gray line), as a function of the degree of active coupling. The experimental MSM value is placed on the curve to extract the degree of active response of the right cell in the experiment. (**D**) Contour analysis of the free stress fiber. In the experiment, the distance between the free fibers as a function of x is measured, as shown in the image on the left. An example for a contour model simulations is shown in the right. (**E**) The contour strain after photoactivation is calculated from the distance measurements shown in (**D**) by dividing the distance between the free stress fibers for each point in x-direction after and before photoactivation. Similarly to the FEM simulation, in the contour simulation, the right half of the contour is progressively activated to obtain the curve family shown in the right plot. (**F**) Response of the right half (normalized by the total response), obtained from the model (gray line), as a function of the degree of active coupling. The experimental strain value is placed on the curve to extract the degree of active response of the right cell in the experiment. (**G**) A cartoon showing our interpretation of the results shown in panels (**A–F**). The traction force analysis only measures forces that are transmitted to the substrate, which are dominated by the activity of the stress fibers. The contour of the free fiber is determined by the activity of the actin cortex and the free stress fiber. Thus, the strong active coupling in the contour suggests strong active coupling of the cortices and the comparatively weak active coupling of the forces suggests a weak active coupling of the stress fibers. The figure shows data from n = 17 doublets from N = 2 samples. All scale bars are $10\,\mu m$ long.

Interestingly, this analysis showed different coupling behaviors in the x- and y-directions. We found positive active coupling in the y-direction (0.2), but negative coupling in the x-direction (–0.05) (yellow square, *Figure 4C*). This may be because all forces in y-direction are balanced between the cell and the substrate, but not across the junction. This signifies that each cell can contract independently from one another in this direction. In contrast, the forces in the x-direction must always be balanced by interaction between the cells across the junction, similar to a 'tug of war.'

To test our hypothesis of independent contraction in the y-direction, we measured the distance between the free stress fibers along the x-axis (*Figure 4D*) to get a readout for cortical tensions not transmitted to the substrate. The ratio of the inter-stress fiber distance during and before photoactivation defines a contour strain along the x-direction (*Figure 4E*). We compared experimental contour strain to the contour strain in simulations, in which we again progressively activated the right half of the contour (*Figure 4D and E*) and repeated the same analysis as in *Figure 4C*. We found a degree of coupling of 0.8, indicating a global active contraction of the unstimulated cell (*Figure 4F*). This is consistent with the active positive coupling measured in the y-direction using MSM (*Figure 4C*). Overall both TFM and surface tension analyses showed active coupling between the two regions. However, active coupling was weaker in TFM measurements, perhaps because the cortices of the two cells are more strongly actively coupled than the stress fibers.

In conclusion, traction forces, as measured by TFM, show weaker active coupling between activated and nonactivated region than cortical tensions, as inferred by measurement of contour strain. The traction forces are dominated by the activity of the stress fibers, both internal and on the periphery, because most forces are found in the corners of the doublet. The only area where the cortex can transmit forces to the substrate is along the vertical fiber in horizontal direction. If this force were substantial, it should point much more horizontally and be much more constant, without the strong hotspots in the corners. The contour of the free fiber, on the other hand, is determined by the activity of the actin cortex and the free stress fiber. Thus, contour analysis suggests strong active coupling of the cortices and the comparatively weaker active coupling observed in cellular stress distributions may occur because internal stress fibers are coupled to the substrate and transmit little stress across the cell junction (*Figure 4G*).

## Mechanical stresses transmit most efficiently perpendicularly to the axis of mechanical and structural polarization in doublets

Our data indicated that active coupling of contractions in the y-direction is much higher in doublets than in the x-direction. We hypothesized that active coupling may be modulated by mechanical and structural polarization of the cells. To test this, we sought to vary structural and mechanical polarization of doublets by changing the aspect ratio of the underlying micropatterns from 1 to 2, 1 to 1, and to 2 to 1 (y to x ratio) while maintaining a constant spreading area. Mechanical polarization and structural polarization were quantified as previously. We found that structural and mechanical polarization are tightly correlated and vary greatly in between the three different aspect ratios (*Figure 5A–C*). For example, on micropatterns with 1 to 2 aspect ratio, both stress fibers and force patterns were oriented horizontally whereas on 2 to 1 they were oriented vertically.

Next, we examined the link between structural polarization and stress transmission. For each aspect ratio, we repeated the local activation experiments (*Figure 4*, *Figure 5D–F*, Animation 6). These optogenetically induced stresses transmit from the sender cell to the receiver cell, that is, from left to right. We observed markedly different behavior depending on aspect ratio. In 1 to 2 doublets, cells are polarized mechanically and structurally along the direction of stress transmission and, after activation of left hand cell, the right cell reacts by relaxing. In contrast, in 2 to 1 doublets, cells are polarized mechanically and structurally perpendicular to the direction of stress transmission and activation of the left-hand cell leads to contraction of the right-hand cell. We then computed the degree of active coupling as previously and found that the degree of active coupling increased with increasing mechanical and structural polarization (*Figure 5D–G*).

We then investigated whether a similar effect could be observed for cortical tensions and performed the contour analysis as in *Figure 4*. Here we saw, in agreement with *Figure 4E*, that the contour deformation is very symmetrical in both the 1 to 1 and the 2 to 1 doublets, but much less in the 1 to 2 doublets, where the degree of active coupling is lower. The quantification of the degree of active coupling here is lower for the 2 to 1 than for the 1 to 1, but the uncertainty of this

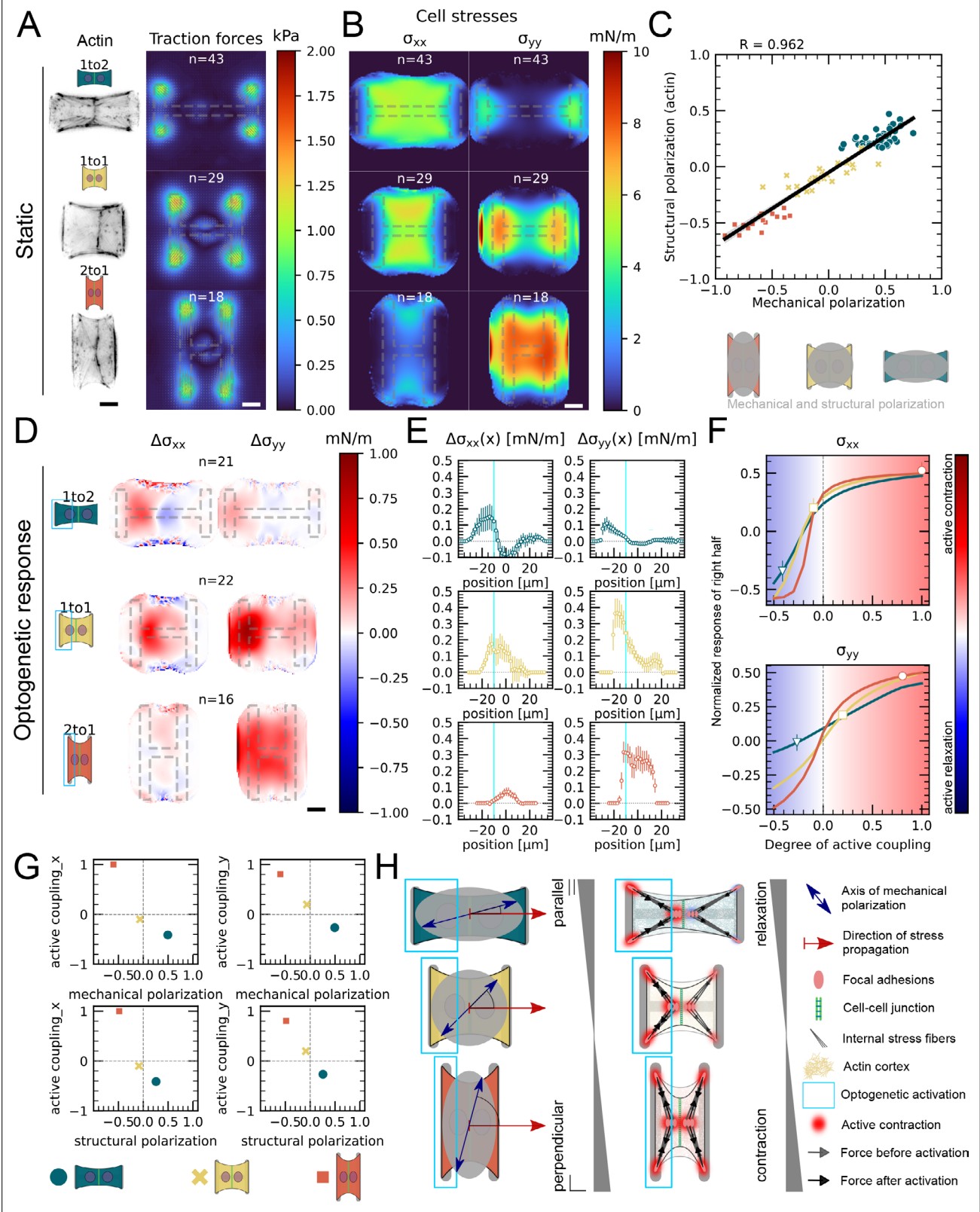

**Figure 5.** Mechanical stresses transmit most efficiently perpendicularly to the axis of mechanical and structural polarization in doublets. (**A**) Actin images (left) and average traction stress and force maps (right) of cell doublets on H-patterns with different aspect ratios (1 to 2, 1 to 1, and 2 to 1). (**B**) Average cell stress maps calculated by applying a monolayer stress microscopy algorithm to the traction stress maps. (**C**) Correlation plot of mechanical and structural polarization. The black line shows the linear regression of the data, and the shaded area shows the 95% confidence interval for this regression.

*Figure 5 continued on next page*

*Figure 5 continued*

The R-value shown corresponds to the Pearson correlation coefficient. (**D**) Stress maps of the difference of xx-stress (left) and yy-stress (right) before and after photoactivation. (**E**) Average over the y-axis of the maps in (**D**). Data is shown as circles with the mean ± SEM. (**F**) Response of the right half (normalized by the total response), obtained from the model (gray line), as a function of the degree of active coupling. The experimental monolayer stress microscopy (MSM) value is placed on the curve to extract the degree of active response of the right cell in the experiment. All scale bars are $10\,\mu m$ long. (**G**) The degree of active coupling plotted against the average mechanical and structural polarization. (**H**) A cartoon showing our interpretation of the data shown in panels (**A–F**). The relative response of the right cell in response to the activation of the left cell varies strongly in the different aspect ratios. In the 1 to 2 doublet, where polarization and transmission direction are aligned, the right cell relaxes, whereas in the 2 to 1 doublet, where the polarization axis is perpendicular to the transmission direction, the right cell contracts almost as strongly as the left cell. The figure shows data from n = 43 1 to 2 doublets from N = 6 samples, n = 29 1 to 1 doublets from N = 2 samples, and n = 18 2 to 1 doublets from N = 3 samples. For the analysis of the optogenetic data, doublets with unstable stress behavior before photoactivation were excluded. All scale bars are $10\,\mu m$ long.

The online version of this article includes the following figure supplement(s) for figure 5:

**Figure supplement 1.** Contour strain measurements of doublets with varying aspect ratios.

quantification is quite high because the contour strain is small, so this is likely due to the noise in the strain measurements (*Figure 5—figure supplement 1*). Altogether, we conclude that mechanical stresses transmit most efficiently perpendicularly to the axis of mechanical and structural polarization in doublets (*Figure 5G*).

## Mechanical stresses transmit most efficiently perpendicularly to the axis of mechanical and structural polarization in small cell clusters

Finally, we investigated whether this conclusion is generalizable to larger systems. Because it is very challenging to position three or four cells on appropriate patterns, we turned to small monolayers. We confined about 10–20 cells on $150\,\mu m \times 40\,\mu m$ rectangular micropatterns. We again performed TFM and MSM experiments as well as live imaging of F-actin and quantified the mechanical and structural polarization for micropatterns with aspect ratios of 1:4. We observed prominent actin cables at the periphery of the small monolayers with less marked stress fibers internally. In these conditions, the tissue is mechanically and structurally polarized along the long axis of the pattern (*Figure 6A–C*, *Figure 6—figure supplement 1*).

We then characterized the efficiency of stress propagation parallel and perpendicular to the axis of tissue polarization. To this end, we photoactivated either the top half or the left half of the tissues. In our experiments, we observed again an increase in traction forces and cell stress both in the activated and in the nonactivated region. We computed the degree of active coupling in the same way as for doublets using our FEM model and found that active coupling is higher, when the direction of stress propagation is perpendicular to the axis of mechanical and structural polarization of the tissue. Additionally, we measured the distance $d$ over which the stress attenuates to 20% of its maximum and found that $d$ is, on average, threefold larger when the direction of stress propagation is perpendicular to the axis of polarization (*Figure 6D–F*). We conclude the correlation between mechanostructural polarization and active coupling observed in doublets is also present in larger groups of cells. In summary, active coupling and its correlation with mechanical and structural polarization seem to be typical for epithelia, independent of size (*Figure 6G*).

## Discussion

Intercellular forces play a major role in regulating and coordinating tissue morphogenesis. Recent work has shown that mechanical forces participate in long-range signaling, propagating over large distances at which they can be received and interpreted by other cells (*Vishwakarma et al., 2018*). However, we have little quantitative insight of how cell-generated forces propagate across intercellular junctions or which cellular structures

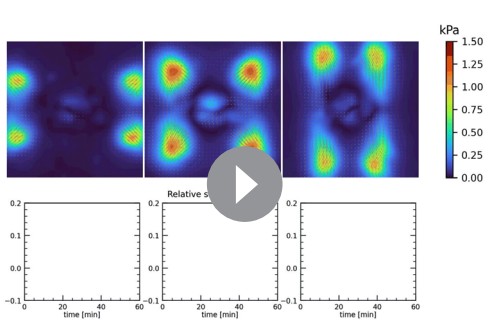

**Animation 6.** Average traction force maps (top) and relative strain energy divided in left (bright) and right (dark) half (right) over time of locally photoactivated 1 to 2 (blue), 1 to 1 (yellow), and 2 to 1 doublets (red).

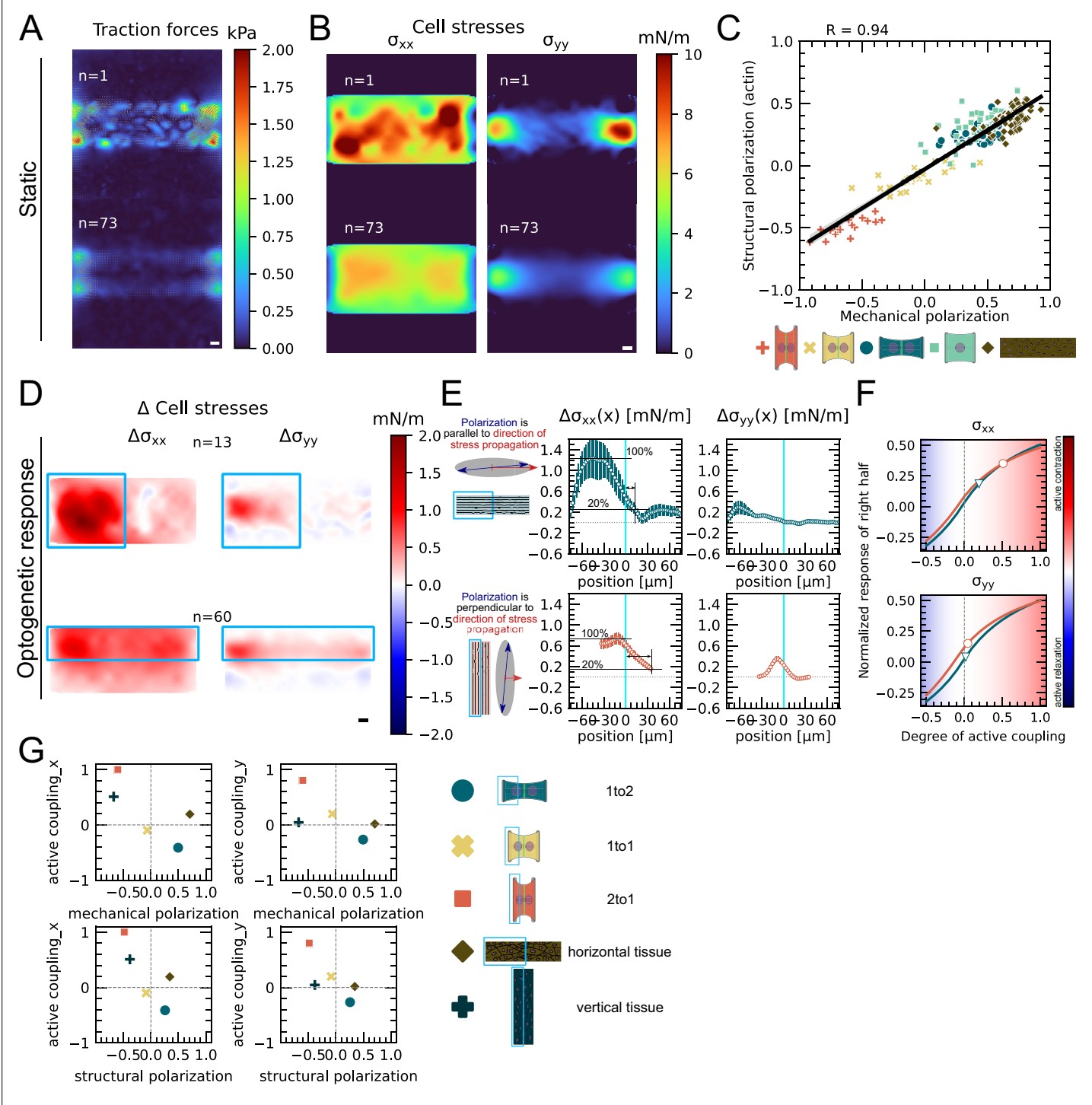

**Figure 6.** Mechanical stresses transmit most efficiently perpendicularly to the axis of mechanical and structural polarization in small monolayers. (**A**) Representative (top) and average (bottom) maps of traction forces and stresses of a small monolayer on rectangular micropattern. (**B**) Representative example and average cell stress maps calculated by applying a monolayer stress microscopy algorithm to the traction stress maps. (**C**) Correlation plot of mechanical and structural polarization across all conditions. The black line shows the linear regression of the data, and the shaded area shows the 95% confidence interval for this regression. The R-value shown corresponds to the Pearson correlation coefficient. (**D**) Stress maps of the difference of xx-stress (left) and yy-stress (right) before and after photoactivation. (**E**) Average over the y-axis of the maps in (**D**). Data is shown as circles with the mean ± SEM. (**F**) Response of the right half (normalized by the total response), obtained from the model (gray line), as a function of the degree of active coupling. The experimental monolayer stress microscopy (MSM) value is placed on the curve to extract the degree of active response of the right cell in the experiment. (**G**) The degree of active coupling plotted against the average mechanical and structural polarization. All scale bars are $10\,\mu$m long. The

*Figure 6 continued on next page*

*Figure 6 continued*

figure shows data from n = 13 tissues from N = 2 samples photoactivated on the left and from n = 60 tissues from N = 3 samples photoactivated on the top. All scale bars are $10\,\mu$m long.

The online version of this article includes the following figure supplement(s) for figure 6:

**Figure supplement 1.** Phalloidin stainings of actin structures of small tissues.

modulate propagation. This is because direct measurement of intercellular forces and cell internal stresses within embryos or tissues is very challenging and most of our knowledge of the distribution of these forces is inferred from theoretical models (*Roffay et al., 2021*; *Yang et al., 2017*; *Saias et al., 2015*; *Brodland et al., 2010*). By combining quantitative measurements of cellular stresses and cell shape with optogenetic control of contractility and mathematical modeling, here we showed that force signal propagation within cellular assemblies is an active process whose amplification mechanism is controlled by the mechano-structural polarization of the system.

Our results revealed the presence of active coupling between cells. This was demonstrated before by *Liu et al., 2010*, but a thorough quantification of this active coupling has been lacking. Photoactivation of one cell in our doublet leads to contraction, sending a force signal. The receiver cell reacts to this signal with an active contractile response. We quantified the response of the receiver cell by comparing experimental traction force and cell stress data with an FEM model. We found that a purely passive reaction of the receiver cell cannot account for the data and therefore concluded that the receiver cell reacts actively. This active coupling mechanism increases the spatial range a mechanical signal can travel to about one or two cell lengths according to our data. Furthermore, analysis of the cell shape showed very high symmetry of shape deformation despite the asymmetrical photoactivation. This shape deformation is dominated by the activity of the actin cortex, and a comparison of this measurement with a mathematical contour model led us to conclude that the active coupling of the cortices is stronger than that of stress fibers. However, this is probably strongly influenced by tissue and cell mechanical properties and by geometry and mechanical properties of the substrate. Additionally, we tested only transient signals. Maintaining signal strength over longer periods of time could also lead to farther transmission of force signals. Compared to chemical signals, these mechanical signals can travel very fast: indeed, with our temporal resolution of one frame per minute, no delay between receiver and sender cell was apparent. In contrast, when we carried out the same activation protocol on a single cell that had the same area as the doublets, the nonactivated region displayed acute fluidization of the actin structure. Thus, in the absence of an intercellular junction, localized contraction leads to actin flow instead of the stress buildup observed in doublets. Therefore, cellularization of the tissue may allow compartmentalization of stress and efficient transmission of stress, allowing the tissue to act as an elastic material rather than a viscous fluid.

Several subcellular features determined the efficiency of active coupling. Indeed, our experiments revealed that intercellular coupling strongly depends on the anisotropy of F-actin organization and force distribution. We found that the magnitude of contraction of the receiving cells relative to the sender cells depends on the direction and magnitude of its mechano-structural polarization. If the tissue's or doublet's polarization axis is perpendicular to the axis between sender and receiver cells, the receiver cells react more strongly and the signal travels farther. However, determining the exact contribution of subcellular structures remains challenging because the cell forms a highly coupled system comprising dynamic mechanotransduction feedback loops. Future work will be necessary to determine the molecular mechanisms detecting the mechanical signal, transducing it, and amplifying it. In particular, it will be interesting to investigate how the active contraction of the receiver cell depends on its own mechano-structural polarization and that of the sender cell. Currently, the nature of the stimulus detected by the receiver cell is unclear. We note that mechanics and biochemistry are closely coupled because strain can change biochemistry by changing concentrations and spatial localization, and stress on single molecules can open cryptic binding sites or increase dissociation constants. One important element that could be studied in future work is the role of E-cadherin in this active coupling, which is highly likely to be important for this process. Knockdown or overexpression studies, although technically challenging, could give important clues to understand the molecular mechanisms behind the active coupling between cells.

Finally, our study of epithelial monolayers shows that the supracellular organization of actin is a major regulator of force propagation within tissues. Forces are transmitted more efficiently in a

direction perpendicular to the axis of actin polarity also in small monolayers. These results give rise to several interesting conclusions. First, recent studies have proposed that groups of cells can behave as a 'supracellular unit,' which share many of the characteristics of the individual cells that it consists of (*Vedula et al., 2013*; *Khalilgharibi et al., 2019*; *Shellard et al., 2018*). Some emerging mesoscale phenomena, such as collective gradient sensing, might be explained by common principles, such as supracellular polarity and supracellular force transmission (*Sunyer et al., 2016*; *Tambe et al., 2011*; *Trepat et al., 2009*; *Vedula et al., 2014*; *van Helvert et al., 2018*). Our findings complement those results, as we show that the correlation between mechano-structural polarization and force signal transmission distance holds true across scales. Second, at a much larger scale, we speculate that propagation through active coupling may have important implications in developmental processes, such as convergent extension in the *Xenopus* mesoderm. In these tissues, cells are planary polarized in a direction perpendicular to the extension of the tissue, and the convergence and extension of the tissue are driven by directed contraction and migration of the cells (*Wallingford et al., 2000*). Our results suggest that preferential transmission of active contraction perpendicular to the polarization axis of the cells could amplify this mechanism and contribute to the robustness of the process.

## Materials and methods
### Cell culture
Opto-MDCK and opto-MDCK LifeAct cells have been kindly provided by Manasi Kelkar and Guillaume Charras. Both cell lines were cultured at 37°C and in 5% $CO_2$ atmosphere in DMEM (Life Technologies) medium containing 10% heat-inactivated FBS (Life Technologies) and 1% penicillin/streptomycin (Sigma-Aldrich). Between 20,000 and 50,000 cells were plated on the micropatterned hydrogels. After 1 hr, cells were checked for their adhesion to the hydrogels. In case of excessive amount of cells, the sample was rinsed with fresh medium to wash off the nonadhered cells. Cells were let spread on patterns for 16–28 hr. Data from timelapse experiments (not shown here) showed that on average most doublets seen on the sample at this point have started as single cells and divided on the pattern to form a doublet. However, we did not control for this, so it is possible that some of the doublets in this study were two different cells to begin with. Some timelapses of forming doublets are shown in Animation 7 and Animation 8. Cells were checked for mycoplasm contamination and tested negative.

### Cell fixing and immunostaining
First, cells were fixed for 10 min with 4% PFA diluted in PBS. Next, the cell membrane was permeabilized with 0.5% Triton X-100 for 5 min. Cells were then washed twice with TBS and blocked at room temperature for 1 hr with a blocking buffer solution containing TBS, 1% bovine serum albumin (BSA, Sigma-Aldrich), and 50 mM glycine (Sigma-Aldrich). Then, cells were incubated for 2 hr in a dilution of primary antibodies with blocking buffer. For E-cadherin stainings, a 1:200 dilution of DECMA-1 (Thermo Fisher 14-3249-82) was used and for vinculin stainings a 1:400 dilution of hVIN-1 (Sigma-Aldrich V9131) was used. Cells were then washed three times with TBS for 10 min each. Then cells were incubated in a dilution of secondary antibodies, Alexa 555-conjugated phalloidin and DAPI in blocking buffer. For E-cadherin stainings, a 1:1000 dilution of Alexa 647-conjugated anti-rat (Sigma-Aldrich SAB4600186) was used; for vinculin stainings, a 1:1000 dilution of Alexa 647-conjugated anti-mouse (Thermo Fisher A-21235) and a 1:1000

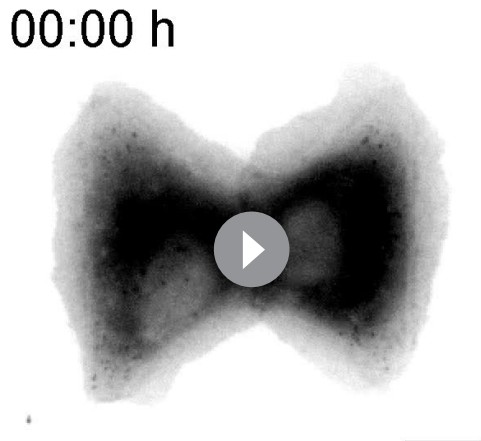

**00:00 h**

**10 μm**

**Animation 7.** Cry2 distribution with photoactivation of left cell in a doublet with increasing power densities. First pulse: $0.18\,\mathrm{mW\,mm^{-2}}$; second pulse: $0.9\,\mathrm{mW\,mm^{-2}}$; third pulse: $1.8\,\mathrm{mW\,mm^{-2}}$; fourth pulse: $3.6\,\mathrm{mW\,mm^{-2}}$; fifth pulse: $9\,\mathrm{mW\,mm^{-2}}$; sixth pulse: $18\,\mathrm{mW\,mm^{-2}}$.

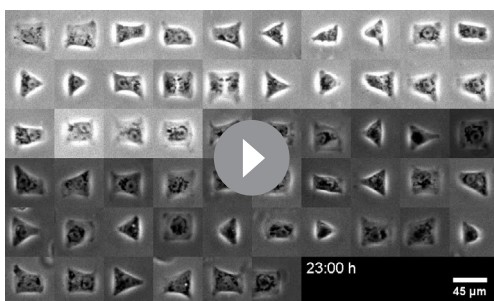

**Animation 8.** Cropped images from brightfield timelapse of doublets forming on H-patterns.

dilution for phalloidin and DAPI. Fixed cells were then mounted with Mowiol 4-88 (Polysciences, Inc) onto glass slides and kept at 4°C until imaging.

## Preparation of micropatterned polyacrylamide gels

Patterned PAA hydrogels were prepared according to the glass method described previously in *Vignaud et al., 2014*. In short, $32\,\mathrm{mm}$ coverslips were first plasma cleaned for $60\,\mathrm{s}$ and then incubated with a drop of PLL-PEG $0.1\,\mathrm{mg\,mL^{-1}}$ in HEPES $10\,\mathrm{mM}$, pH 7.4 for $30\,\mathrm{min}$ at room temperature. Then, coverslips were rinsed with a squirt bottle of MilliQ water and carefully dried with a nitrogen gun. The coverslips were then placed on a quartz photomask (Toppan) on a $10\,\mu\mathrm{L}$ drop of MilliQ water. Excess water was removed by placing a kimwipe on the coverslips, a flat surface on top (e.g. the lid of a petridish) and then pressing gently. The coverslips on the photomask were then exposed to deep-UV for $5\,\mathrm{min}$. After recovery from the photomasks, the coverslips are incubated with $20\,\mathrm{g\,mL^{-1}}$ fibronectin (Sigma-Aldrich) and $20\,\mathrm{g\,mL^{-1}}$ Alexa 488-conjugated fibrinogen (Invitrogen) in $100\,\mathrm{mM}$ sodium bicarbonate buffer for $30\,\mathrm{min}$ at room temperature. To prepare the gels, a $47\,\mu\mathrm{L}$ drop of $20\,\mathrm{kPa}$ mix of polyacrylamide and bis-acrylamide (Sigma-Aldrich) was prepared (see *Tse and Engler, 2010* for the proportions). To perform TFM, carboxylate-modified polystyrene fluorescent microbeads (Invitrogen F-8807) were added to the polyacrylamide premix and sonicated for $3\,\mathrm{min}$ to break bead aggregates. A second coverslip of the same size is then placed on top, after previous silanization with a solution of $5\,\mathrm{mL}$ 100% ethanol, $18.5\,\mu\mathrm{L}$ Bind Silane (GE Healthcare Life Sciences) and $161\,\mu\mathrm{L}$ 10% acetic acid (Sigma-Aldrich) for $5\,\mathrm{min}$. During the polymerization process, the hydrogel adheres to the silanized coverslip and fibronectin proteins are trapped within the polyacrylamide mesh. The silanized coverslip is finally detached by wetting it with MilliQ water, letting the gel rehydrate for $5\,\mathrm{min}$, and lifting it up with a scalpel. Hydrogels were stored in $100\,\mathrm{mM}$ sodium bicarbonate buffer at 4°C for maximum 2 d before cell seeding.

## Imaging and optogenetic photoactivation

All experiments were conducted 16–28 hr after seeding the cells on the sample. Then the cells were observed on an inverted Nikon Ti-E2 microscope with an Orca Flash 4.0 sCMOS camera (Hamamatsu), a temperature control system set at 37°C, a humidifier, and a $CO_2$ controller. For the opto-experiments on cell doublets and singlets, a Nikon ×60 oil objective was used and for the opto-experiments on tissues a Nikon ×40 air objective was used. The E-cadherin and vinculin staining images were taken with an Eclipse Ti inverted confocal microscope (Nikon France Instruments, Champigny sur Marne, France), equipped with sCMOS prime camera (Photometrics), a ×60 objective, and a CSU X1 spinning disk (Yokogawa, Roper Scientific, Lisses, France). MetaMorph software was used for controlling the microscope (Universal Imaging Corporation, Roper Scientific, Lisses, France). Unless otherwise stated, all photoactivations were done with one pulse per min for $10\,\mathrm{min}$, and each pulse had a duration of $200\,\mathrm{ms}$, a power density of $0.9\,\mathrm{mW\,mm^{-2}}$, and a wavelength of $470\,\mathrm{nm}$. The power density was measured with a power meter right after the objective by shining light on a surface of a given size and dividing the measured power by this size. Photoactivation regions were aligned with respect to the micropattern to ensure reproducibility.

## Traction force microscopy and monolayer stress microscopy

Force measurements were performed using a method described previously (*Tseng et al., 2011*). In short, fluorescent beads were embedded in a polyacrylamide substrate with 20 kPa rigidity and images of those beads were taken before, during, and after photoactivation. At the end of the experiment, cells were removed with 2.5% Trypsin and an unstressed reference image of the beads was taken. The displacement field analysis was done using a homemade algorithm based on the combination of particle image velocimetry and single-particle tracking. After correcting for experimental drift, bead images were divided into smaller subimages of $13.8\,\mu\mathrm{m}$ width. The displacement between

corresponding bead subimages was obtained by cross-correlation. After shifting the stressed subimages to correct for this displacements, the window size is divided by 2 and new displacement values are determined by cross-correlations on the smaller subimages. This procedure is repeated twice. On the final subimages, single-particle tracking was performed: this ensures that the displacement measurement has the best possible spatial resolution at a given bead density. Erroneous vectors were detected by calculating the vector difference of each vector with the surrounding vectors. If the vector magnitude was higher than $2.5\,\mu\mathrm{m}$ or the vector difference higher than $1\,\mu\mathrm{m}$, the vector was discarded and replaced by the mean value of the neighboring vectors. Only the first frame of each movie was compared to the unstressed reference image. All subsequent frames were compared to their predecessor. This leads to more precise measurements because the displacements are much smaller. From the bead displacement measurements, a displacement field was then interpolated on a regular grid with $1.3\,\mu\mathrm{m}$ spacing. Cellular traction forces were calculated using Fourier transform traction cytometry with zero-order regularization (*Milloud et al., 2017*; *Sabass et al., 2008*) under the assumption that the substrate is a linear elastic half-space and considering only displacement and stress tangential to the substrate. To calculate the strain energy stored in the substrate, the scalar product of the stress and displacement vector fields was integrated over the surface of the whole cell. The algorithm was implemented in MATLAB and is available in *Ruppel et al., 2023*. For the contour model, $\sigma_\mathrm{x}$ was measured on the TFM maps by summing up the x-traction stresses in a window around the center of the vertical fiber. Within the MSM framework, cell internal stresses were calculated from the traction stress with the code from *Bauer et al., 2021*. To do this calculation, the cell is assumed to behave like a thin, elastic sheet that is attached to a substrate and then contracts. Equilibrium shape is reached, when the active stress that leads to the contraction is balanced by the elastic stress that builds up within the sheet and in the substrate. The resulting stress is the sum of the active and the passive stress in the elastic sheet and is independent of its elastic modulus.

## Estimation of active coupling in doublets and tissues

In order to estimate the active coupling present in the doublets and tissues, we performed simulations for different levels of active coupling, ranging from –1 to 1, and calculated stress maps. We then subtracted the stress maps obtained before photoactivation from those obtained after photoactivation. These results were averaged along the y-axis, and the resulting curves are plotted in *Figure 4B*. To compare with experimental data, we also calculated the corresponding curves in the experiment using stress maps obtained through MSM, which are plotted in *Figures 4B–6E*. We repeated this process for the contour model and contour strain, yielding the plot in *Figure 4E*. To compare the theoretical and experimental curves, we normalized the stress response of the right half by integrating the area under each curve in the right half and dividing it by the total area. We then plotted the resulting normalized stress response against the degree of active coupling in the simulation, yielding the plots in *Figures 4C, F, 5C, 6F*, and *Figure 5—figure supplement 1C*. Finally, we compared these plots with the normalized stress and strain responses from the experiment by placing the experimental value on the theoretical curve and reading the corresponding degree of active coupling on the x-axis.

## Fiber tracking

A semi-automatic procedure was used to detect and track the actin fibers at the cell contour over time. First, the operator clicks on the endpoints of each fiber on the first image of a timelapse. The adherent fibers are very static and straight, so, in this case, we just draw a straight line between the two end points. The free fibers are curved and move over time. To follow the shape of a given fiber over time, we used a custom script: on each image, parallel line profiles are drawn at regular intervals in between the two defined endpoints, in a direction perpendicular to the overall fiber direction; each profile is analyzed to detect the point where it intersects the fiber using intensity variation as criterion. The line linking these points describes the actin fiber position at each time point. In order to filter out badly detected points, the consistency of the resulting positions is analyzed over both time and space. Temporal filtering consists of first a median filter over five time points and the removal of outliers. Within a moving time window of 10 time points, positions distant from the average value by more than two times the standard deviation are deleted. Spatial filtering includes also removal of outliers, defined as being distant from the spatial average position by more than three times the standard deviation. Then the angle of lines joining adjacent points is computed at each position and badly tracked

points are excluded by ensuring that these angles stay below 15°. Finally, we use this tracking data to create a stack of masks for each cell which accurately describes the complete contour of the cell. The algorithm was implemented in MATLAB and is available in *Ruppel et al., 2023*.

## Actin polarization analysis

To measure the average polarization of the internal actin network, we analyze the orientation of the internal actin network using the structure tensor formalism (*Jähne, 1995*). For each pixel with intensity $I(x, y)$, the structure tensor $J$ is calculated over a Gaussian local neighborhood $w(x, y)$ with a waist of 3 pixels, according to *Equation (1)*.

$$
\begin{aligned}
J_{11} &= \iint w(x, y) \left( \frac{\partial I(x, y)}{\partial x} \right)^2 dx\, dy \\
J_{22} &= \iint w(x, y) \left( \frac{\partial I(x, y)}{\partial y} \right)^2 dx\, dy \\
J_{12} = J_{21} &= \iint w(x, y) \left( \frac{\partial I(x, y)}{\partial x} \right) \left( \frac{\partial I(x, y)}{\partial y} \right) dx\, dy
\end{aligned}
\tag{1}
$$

The orientation angle $\theta$ on this local neighborhood corresponds to the direction of the main eigenvector of the structure tensor and is obtained by *Equation (2)*.

$$
\tan(2\theta) = \frac{2J_{12}}{J_{22} - J_{11}}
\tag{2}
$$

This angle is only meaningful if the image shows oriented structures in this neighborhood. This confidence can be estimated from the coherency, which quantifies the degree of anisotropy and is calculated from the structure tensor according to *Equation (3)*. Values with a coherency value under 0.4 were excluded before averaging the orientation angles over the cell to obtain the mean direction of the actin network. The degree of polarization is then obtained according to *Equation (4)*. The algorithm was implemented in MATLAB and was used before by *Mandal et al., 2014*. The version used for this work is available in *Ruppel et al., 2023*.

$$
Coherency = \frac{\sqrt{(J_{22} - J_{11})^2 + 4J_{12}^2}}{J_{11} + J_{22}}
\tag{3}
$$

$$
Polarization = <\cos(2(\theta - \theta_{mean}))>
\tag{4}
$$

## Actin intensity measurement

To measure the actin intensity in the left and the right half of the doublet/singlet, we first segment the cells using the masks obtained from the fiber tracking. We reduce its size a little bit to exclude the external stress fibers from the measurement. We then divide the doublet/singlet vertically in two halves and sum up all the intensity values within the region of interest, yielding one intensity value per frame and per half. This intensity over time is then normalized by the intensity value of the average over the first 20 frames before photoactivation.

## Statistical analysis and boxplots

All boxplots show the inner quartile range as boxes and the whiskers extend to 1.5 times the inner quartile range. The notches show the 95% confidence interval for the median, and the white dot shows the sample mean. The Mann–Whitney–Wilcoxon $U$ test was used to test for differences between singlets and doublets, with ns: p>0.05, *p<0.05, **p<0.01,***p<0.001, and ****p<0.0001.

## Data exclusion for optogenetic experiments

Many of the cells showed an unstable baseline energy level, which made it difficult to judge the impact of the optogenetic activation. Thus, we quantified the baseline stability of each cell by applying a linear regression to the relative strain energy curve before photoactivation and excluded all cells with a slope larger in absolute value than a threshold value. For *Figure 3*, this process excluded 16 globally activated doublets, 7 globally activated singlets, 12 locally activated doublets, and 17 locally activated

singlets. For *Figure 5D–F*, this process excluded 22 1 to 2 doublets, 7 1 to 1 doublets, and 2 2 to 1 doublets.

## Acknowledgements

MK was supported by a Swiss National Science Foundation early postdoctoral fellowship (P2LAP3_164919) and by a European Research Council consolidator grant to GC (CoG-647186). GCa acknowledges financial support from the ANR SupraWaves project, grant ANR-19-CE13-0028. TB acknowledges funding through CNRS grants (Actions Interdisciplinaires 2017, DEFI Instrumentation aux limites 2017, Tremplin@INP 2021, PEPS CNRS-INSIS 2021). USS acknowledges funding through a Deutsche Forschungsgemeinschaft (DFG, MechanoSwitch project SCHW 834/2-1). MB acknowledges financial support from the French Agence Nationale de la Recherche (ANR) MechanoSwitch project, grant ANR-17-CE30-0032-01. This work was supported by the Center of Excellence of Multifunctional Architectured Materials 'CEMAM' (no. AN-10-LABX-44-01).

## Additional information

### Funding

| Funder | Grant reference number | Author |
| --- | --- | --- |
| Swiss National Science Foundation | P2LAP3 164919 | Manasi Kelkar |
| European Research Council | CoG-647186 | Guillaume Charras |
| Agence Nationale de la Recherche | ANR-19-CE13-0028 | Giovanni Cappello |
| Centre National de la Recherche Scientifique | Actions Interdisciplinaires 2017 | Thomas Boudou |
| Deutsche Forschungsgemeinschaft | SCHW 834/2-1 | Ulrich S Schwarz |
| Agence Nationale de la Recherche | ANR-17-CE30-0032-01 | Martial Balland |
| Centre National de la Recherche Scientifique | DEFI Instrumentationaux limites 2017 | Thomas Boudou |
| Centre National de la Recherche Scientifique | Tremplin@INP 2021 | Thomas Boudou |
| Centre National de la Recherche Scientifique | PEPS CNRS-INSIS 2021 | Thomas Boudou |

The funders had no role in study design, data collection and interpretation, or the decision to submit the work for publication.

### Author contributions

Artur Ruppel, Dennis Wörthmüller, Conceptualization, Data curation, Software, Formal analysis, Validation, Investigation, Visualization, Methodology, Writing – original draft, Writing – review and editing; Vladimir Misiak, Data curation, Formal analysis, Validation, Investigation, Visualization, Methodology; Manasi Kelkar, Resources, Validation, Investigation, Methodology; Irène Wang, Data curation, Software, Formal analysis, Validation, Investigation, Visualization, Methodology; Philippe Moreau, Adrien Méry, Resources, Investigation, Methodology; Jean Révilloud, Resources; Guillaume Charras, Conceptualization, Supervision, Funding acquisition, Validation, Investigation, Methodology, Writing – original draft, Writing – review and editing; Giovanni Cappello, Thomas Boudou, Conceptualization, Supervision, Funding acquisition, Validation, Investigation, Methodology, Writing – review and editing; Ulrich S Schwarz, Martial Balland, Conceptualization, Resources, Data curation, Software, Formal analysis, Supervision, Funding acquisition, Validation, Investigation, Visualization, Methodology, Writing – original draft, Project administration, Writing – review and editing

**Author ORCIDs**
Artur Ruppel https://orcid.org/0000-0002-2541-8257
Dennis Wörthmüller http://orcid.org/0000-0001-6443-0112
Vladimir Misiak http://orcid.org/0000-0001-6637-8071
Adrien Méry http://orcid.org/0000-0001-9582-0519
Guillaume Charras https://orcid.org/0000-0002-7902-0279
Giovanni Cappello http://orcid.org/0000-0002-5012-367X
Thomas Boudou http://orcid.org/0000-0002-6821-2937
Ulrich S Schwarz https://orcid.org/0000-0003-1483-640X
Martial Balland http://orcid.org/0000-0002-6585-9735

**Decision letter and Author response**
Decision letter https://doi.org/10.7554/eLife.83588.sa1
Author response https://doi.org/10.7554/eLife.83588.sa2

## Additional files

### Supplementary files
• MDAR checklist

### Data availability
All data has been deposited on Dryad. All code has been deposited on Github (copy archived at *Ruppel et al., 2023*).

The following dataset was generated:

| Author(s) | Year | Dataset title | Dataset URL | Database and Identifier |
|---|---|---|---|---|
| Ruppel A | 2023 | Force propagation between epithelial cell doublets | https://dx.doi.org/10.5061/dryad.sj3tx9683 | Dryad Digital Repository, 10.5061/dryad.sj3tx9683 |

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

## Appendix 1

### Overview

Here we present two modeling frameworks that allow us to complement the experimental measurements in different ways. On the one hand, we use FEM simulations of a two-dimensional continuum model to complement the quantification of TFM and MSM. On the other hand, we use a contour model for cellular adhesion to combine TFM and cell shape measurements. The theory supplement is accordingly divided into two sections. In the section "Two-dimensional continuum model", we introduce the continuum model and its parameterization, and briefly describe how we used this modeling approach alongside the experimental data. In the section "Contour model", we focus on the contour model. Here we start with the ATM and explain how we use the underlying theory to translate cell shape measurements into physical quantities. In the second part of this section, we then explain how we solve the central equation of motion in a FEM framework to calculate the shapes of peripheral actin fibers subjected to spatially varying surface tensions. Because both models use FEM methods, we start with the continuum model for which a FEM approach is more standard.

### Two-dimensional continuum model

This mesoscopic model approximates the cell as an elastic continuum. The general constitutive relation can be written as (*Kruse et al., 2005*; *Prost et al., 2015*; *Edwards and Schwarz, 2011*)

$$\sigma_{ij}^{3D} = C_{ijkl}\epsilon_{kl} + \sigma_{ij}^{m,3D}\,,\tag{5}$$

with total stress tensor $\sigma_{ij}^{3D}$, stiffness tensor $C_{ijkl}$, strain tensor $\epsilon_{kl}$, and motor stress tensor $\sigma_{ij}^{m}$. Further, the force balance equation

$$\partial_j\sigma_{ij}^{3D} - b_i = \rho a_i = 0 \tag{6}$$

is used to calculate the deformation of the cell, where $b_i$ is the external body force acting on the cell. For cells or tissues, we always assume the inertial term to vanish.

### Thin-layer approximation

We next assume that the effective thickness of the cell $h_c$ is much smaller than the overall extent of the cell $h_c \ll L_c$. The term effective thickness refers to the thickness of the contractile actomyosin layer coupled to the substrate and not to the full cell, which has variable thickness anyway. Thus, variations along the z-direction are assumed to be small and it is sufficient to consider a thickness-averaged stress tensor given by

$$\tilde{\sigma}^{3D}(x,y) = \frac{1}{h_c}\int_0^{h_c} \mathrm{dz}\,\sigma^{3D}(x,y,z)\,.\tag{7}$$

Averaging the force balance equation leads to a two-dimensional force balance equation in which the thickness-averaged body force is now acting as a traction

$$\frac{1}{h_c}\int_0^{h_c} \mathrm{dz}\,\partial_j\sigma_{ij}^{3D} = \frac{1}{h_c}\int_0^{h_c} \mathrm{dz}\,b_i \tag{8}$$

$$h_c\partial_j\tilde{\sigma}_{ij} = \int_0^{h_c} \mathrm{dz}\,b_i \tag{9}$$

$$\partial_j\sigma_{ij}^{2D} = t_i(x,y)\,.\tag{10}$$

The effective cell thickness is the conversion factor between three-dimensional and two-dimensional quantities, $q^{2D} = q^{3D}h_c$.

### Plane stress

Under plane stress assumption, we set $\sigma_{zz} = \sigma_{xz} = \sigma_{zx} = \sigma_{yz} = \sigma_{zy} = 0$ and further neglect out-of-plane strain $\epsilon_{zz}$. Hooke's law under plane stress conditions can be written in Voigt notation as

$$\begin{bmatrix} \sigma_{xx} \\ \sigma_{yy} \\ \sigma_{xy} \end{bmatrix} = \frac{h_c E_c^{3D}}{1 - \nu_c^2} \begin{bmatrix} 1 & \nu_c & 0 \\ \nu_c & 1 & 0 \\ 0 & 0 & \frac{1 - \nu_c}{2} \end{bmatrix} \begin{bmatrix} \epsilon_{xx} \\ \epsilon_{yy} \\ \epsilon_{xy} \end{bmatrix} . \tag{11}$$

Together with the general version of Hooke's law

$$\sigma_{ij} = \lambda \epsilon_{kk} \delta_{ij} + 2\mu \epsilon_{ij} \tag{12}$$

we determine the 2D Lamé parameter as

$$\lambda = \frac{\nu_c h_c E_c^{3D}}{1 - \nu_c^2}, \quad \mu = \frac{h_c E_c^{3D}}{2(1 + \nu_c)} . \tag{13}$$

## Active Kelvin–Voigt model

The constitutive relation of an active Kelvin–Voigt model in index notation is given by

$$\sigma_{ij} = (1 + \tau_c \frac{\partial}{\partial t})(\lambda \epsilon_{kk} \delta_{ij} + 2\mu \epsilon_{ij}) + \sigma_{ij}^{m} , \tag{14}$$

with stress tensor $\sigma_{ij}$, strain tensor $\epsilon_{ij}$, and the 2D Lamé coefficients as defined in *Equation (13)*. The material relaxation time is defined as $\tau_c = \eta_c/E_c$ with $\eta_c$ denoting the cell viscosity. The linearized strain tensor is defined as

$$\epsilon_{ij} = \frac{1}{2}(\partial_i u_j + \partial_j u_i) , \tag{15}$$

where $u_j$ is the $j$th component of the displacement field vector $\mathbf{u}(\mathbf{x})$. The overall active contraction is described by the anisotropic motor stress tensor $\sigma_{ij}^{m}$ which is split into

$$\sigma_{ij}^{m} = \sigma_{ij}^{bck} + \sigma_{ij}^{opto} , \tag{16}$$

that is, a time-independent background stress to account for the cellular energy baseline level and a time-dependent photoactivation stress tensor describing the stress increase during photo activation (PA).

Based on experimental observations and verification with the MSM analysis of the TFM data, the anisotropy of the cytoskeleton enters the stress tensor for the background stress through the mechanical polarization which is defined as

$$MP = \frac{\sigma_{xx} - \sigma_{yy}}{\sigma_{xx} + \sigma_{yy}} . \tag{17}$$

This leads to

$$\sigma^{bck} = \begin{pmatrix} \sigma_{xx}^{bck} & 0 \\ 0 & \sigma_{yy}^{bck} \end{pmatrix} = \sigma_{xx}^{bck} \begin{pmatrix} 1 & 0 \\ 0 & \frac{1 - MP}{1 + MP} \end{pmatrix} . \tag{18}$$

Upon photoactivation, we assume a time-dependent stress contribution given by

$$\sigma_{opto} = \sigma_{act} \left( 1 - e^{-\frac{t - t_{act}}{\tau_{act}}} \right) \left( 1 - \frac{1}{1 + e^{-\frac{t - \bar{t}}{\tau_{rel}}}} \right) , \tag{19}$$

which is a combination of an increasing saturating exponential and a sigmoidal-shaped decrease (*Appendix 1—figure 1A*).

## Cell–substrate coupling

The cell–substrate coupling is described by *Equation (10)*, where the traction is formulated as

$$\mathbf{t}(\mathbf{x}) = Y(\mathbf{x})\mathbf{u}(\mathbf{x}) \tag{20}$$

which yields

$$\partial_j \sigma_{ij} = Y u_i .$$ (21)

$Y$ denotes the position-dependent spring stiffness density. Combining *Equation (14)* and *Equation (21)*, one can show that the interplay of cellular and substrate elasticity defines a natural length scale (*Edwards and Schwarz, 2011*)

$$l_p^2 = \frac{h_c E_c^{3D}}{Y(1 - \nu_c^2)} ,$$ (22)

known as the force-localization length, which describes how far a point force is transmitted in the elastically coupled isotropic material. According to earlier work (*Banerjee and Marchetti, 2012*; *Mertz et al., 2012*), the spring stiffness density of the substrate can be deduced from the Young's modulus of the substrate $E_s$ via

$$Y = \frac{\pi E_s}{h_{\text{eff}}} ,$$ (23)

in which the effective substrate height is given by an interpolation formula

$$h_{\text{eff}}^{-1} = \frac{1}{h_s 2\pi(1 + \nu_s)} + \frac{1}{L_c} ,$$ (24)

where $h_s$ and $L_c$ denote the substrate height and cell layer size, respectively, and $\nu_s$ is the Poisson's ratio of the substrate. To adapt our theory as close as possible to the traction force computation of the experiments, we assume that the substrate is infinitely thick and therefore we have $h_{\text{eff}} \approx L_c$. Further, we have for the traction forces at the cell–substrate interface

$$\mathbf{T} = Y\mathbf{u} .$$ (25)

The elastic energy stored in the substrate is calculated via

$$U_s = \frac{1}{2} \int_\Omega \mathbf{T}\mathbf{u} \, d\Omega = \frac{1}{2} \int_\Omega Y\mathbf{u}^2 d\Omega .$$ (26)

## Parameterization

Although in principle it is possible to use a downhill-simplex method to find the set of parameters that minimizes the theoretically computed substrate energy against the experimentally measured curve, we nevertheless decide to fix some of the parameters to avoid overfitting. All fixed parameters are listed in *Appendix 1—table 1*. While the substrate parameters are known, we fix the parameters for Young's modulus and viscosity of the cell to typically reported values from the literature (*Edwards and Schwarz, 2011*; *Banerjee and Marchetti, 2012*; *Vishwakarma et al., 2018*; *Hanke et al., 2018*; *Saha et al., 2016*). The fixed substrate parameters yield a spring stiffness density of $Y_s = 1.257 \times 10^9 \, \text{N m}^{-3}$ and a force-localization length of $l_p = 3.25 \, \mu\text{m}$.

## Finite element simulation

We solve the combination of *Equation (14)* and *Equation (21)* for the displacement vector $\mathbf{u}$ of the cell by means of a finite element simulation using the open-source software package FEniCS (*Alnaes et al., 2015*). This approach has been used in several other works (*Edwards and Schwarz, 2011*; *Banerjee and Marchetti, 2012*; *Mertz et al., 2012*; *Oakes et al., 2014*; *Vishwakarma et al., 2018*; *Hanke et al., 2018*; *Solowiej-Wedderburn and Dunlop, 2022*). The full problem statement is given by: find the displacement field vector $\mathbf{u}(\mathbf{x})$ with initial conditions $\mathbf{u}_0 = \mathbf{u}(\mathbf{x}, 0) = 0$ such that together with $\boldsymbol{\sigma} = (1 + \tau_c \frac{\partial}{\partial t})(\lambda \text{tr}(\boldsymbol{\epsilon})\mathbb{1} + 2\mu\boldsymbol{\epsilon}) + \boldsymbol{\sigma}^{\text{m}}$

$$\nabla \cdot \boldsymbol{\sigma} = Y\mathbf{u} \quad \text{in } \Omega \times (0, T]$$ (27)

$$\boldsymbol{\sigma} = 0 \quad \text{on } \partial\Omega \times (0, T] .$$ (28)

Therefore, we derive the weak form of *Equation (21)* by multiplying with a vector-valued test function $\mathbf{v} \in \mathcal{D}(\Omega)$ over the simulation domain $\Omega$. Multiplying *Equation (27)* with the test function and integrating over the whole simulation domain leads to

$$\int_\Omega (\nabla \cdot \boldsymbol{\sigma}) \cdot \mathbf{v} \, d\Omega = \int_\Omega Y\mathbf{u} \cdot \mathbf{v} \, d\Omega. \tag{29}$$

The left-hand side can be integrated using integration by parts, that is, using the following identity

$$\nabla \cdot (\boldsymbol{\sigma}^T \cdot \mathbf{v}) = (\nabla \cdot \boldsymbol{\sigma}) \cdot \mathbf{v} + \boldsymbol{\sigma} : \nabla \mathbf{v}, \tag{30}$$

where we use the standard notation for the inner product between tensors (double contraction) and $\nabla \mathbf{v} = \partial_i (v_j \mathbf{e}_j) \otimes \mathbf{e}_i$ being the vector gradient. *Equation (29)* can be simplified to

$$\int_\Omega \boldsymbol{\sigma} : \nabla \mathbf{v} \, d\Omega - \int_{\partial\Omega} (\boldsymbol{\sigma} \cdot \mathbf{n}) \cdot \mathbf{v} \, d\Gamma + \int_\Omega Y\mathbf{u} \cdot \mathbf{v} \, d\Omega = 0. \tag{31}$$

$\boldsymbol{\sigma} \cdot \mathbf{n}$ is the traction vector at the boundary $\Gamma = \partial\Omega$, which is set to zero in case of stress-free boundaries. We further use that $\boldsymbol{\sigma}$ is symmetric and thus, the double contraction with the antisymmetric part $\mathbf{a}(\mathbf{v}) = \frac{1}{2}(\nabla \mathbf{v} - \nabla \mathbf{v}^T)$ of $\nabla \mathbf{v}$ is zero, that is, $\boldsymbol{\sigma} : \mathbf{a}(\mathbf{v}) = 0$. This allows us to replace $\nabla \mathbf{v}$ by its symmetric part $\mathbf{s}(\mathbf{v}) = \frac{1}{2}(\nabla \mathbf{v} + \nabla \mathbf{v}^T)$ and leads to the final weak form statement

$$\int_\Omega \boldsymbol{\sigma} : \mathbf{s}(\mathbf{v}) \, d\Omega + \int_\Omega Y\mathbf{u} \cdot \mathbf{v} \, d\Omega = 0. \tag{32}$$

Since we are aiming at solving for the displacement vector $\mathbf{u}$, we have to express all terms in the constitutive relation in terms of $\mathbf{u}$

$$\boldsymbol{\sigma} = (1 + \tau_c \frac{\partial}{\partial t})(\lambda \mathrm{tr}(\boldsymbol{\epsilon})\mathbb{1} + 2\mu\boldsymbol{\epsilon}) + \boldsymbol{\sigma}^m \tag{33}$$

$$= \lambda(\nabla \cdot \mathbf{u})\mathbf{I} + \mu(\nabla\mathbf{u} + \nabla\mathbf{u}^T) + \tau_c\lambda(\nabla \cdot \dot{\mathbf{u}})\mathbf{I} + \tau_c\mu(\nabla\dot{\mathbf{u}} + \nabla\dot{\mathbf{u}}^T) + \boldsymbol{\sigma}^m \tag{34}$$

$$= \boldsymbol{\Sigma}_E + \boldsymbol{\Sigma}_{\boldsymbol{\eta}} + \boldsymbol{\sigma}^m. \tag{35}$$

For the time derivatives, we use a backward Euler discretization scheme that is numerically stable even for larger time steps. We set

$$\dot{\mathbf{u}}^{(n+1)} = \frac{\mathbf{u}^{(n+1)} - \mathbf{u}^{(n)}}{\Delta t} \tag{36}$$

and since we are dealing with linear equations the discretization scheme translates directly to

$$\dot{\boldsymbol{\Sigma}}_{E,\eta}^{(n+1)} = \frac{\boldsymbol{\Sigma}_{E,\eta}^{(n+1)} - \boldsymbol{\Sigma}_{E,\eta}^{(n)}}{\Delta t} \tag{37}$$

which enables us to define

$$a(\mathbf{u}^{(n+1)}, \mathbf{v}) = \int_\Omega \boldsymbol{\Sigma}_E^{(n+1)} : \mathbf{s}(\mathbf{v})\Delta t \, d\Omega + \int_\Omega \boldsymbol{\Sigma}_\eta^{(n+1)} : \mathbf{s}(\mathbf{v}) \, d\Omega + \int_\Omega Y\mathbf{u}^{(n+1)} \cdot \mathbf{v}\Delta t \, d\Omega \tag{38}$$

and

$$L^{(n+1)}(\mathbf{v}) = \int_\Omega \boldsymbol{\Sigma}_\eta^{(n)} : \mathbf{s}(\mathbf{v}) \, d\Omega - \int_\Omega \boldsymbol{\sigma}_m : \mathbf{s}(\mathbf{v})\Delta t \, d\Omega \tag{39}$$

after inserting the time-discretized version of *Equation (35)* in *Equation (32)*. Our initial problem statement now reduces to solving

$$a(\mathbf{u}^{(n+1)}, \mathbf{v}) = L^{(n+1)}(\mathbf{v}) \quad \forall \mathbf{v} \in \mathcal{D}(\Omega) \tag{40}$$

*Equation (40)* can be directly handed to the FE solver.

Further, we used the open-source meshing software GMSH (*Geuzaine and Remacle, 2009*) to create a finite element mesh as depicted in *Appendix 1—figure 1B*. We chose an unstrained configuration as initial condition since it is the simplest choice and since we have no experimental access to the actual initial configuration. We checked that this assumption has little effects on

our results. Then we fixed all known parameters in order to match the experimental setup to our simulations. All fixed parameters are gathered in *Appendix 1—table 1* and were fixed throughout the simulations. Next we mathematically defined the pattern geometry of the H-pattern that determines the portion of the simulation domain on which the cell is assumed to establish a connection to the elastic foundation (*Appendix 1—figure 1C*)

$$(x, y)_{Y \neq 0} = \left\{ x, y \,\middle|\, x \leq w - \frac{d}{2} \,\vee\, x \geq \frac{d}{2} - w \,\vee\, -\frac{w}{2} \leq y \leq \frac{w}{2} \right\}. \tag{41}$$

To make our results comparable to other reported values in the literature, we first determined the active background stress by fitting the baseline of strain energy curve (*Equation 26*) to the given experimental substrate strain energy. In a second step, we fitted the temporal evolution of the strain energy by optimizing the free parameters $\sigma_{\text{act}}$, $\tau_{\text{act}}$, $\tau_{\text{rel}}$, and $\tilde{t}$ in Equation (19). The obtained parameters for all fitted conditions are summarized in *Appendix 1—table 2*. The fit results of the doublet and singlet strain energy curves can be seen in *Appendix 1—figure 4D*. At this point, we would like to note that although our model can capture the time course of the strain energy, this plays a minor role for the further evaluation and in comparison with the data. In the course of the project, it turned out that it is sufficient to look at the relative stress increase during photoactivation, that is, the difference between the baseline values and the value at maximum strain energy. Therefore, the curves in the figures are always normalized. Additionally, this allowed us also to use the parameters of the global photoactivation protocol in the context of local photoactivation. Further, we note that at baseline and peak strain energy viscoelasticity does not contribute (extremal value → d/dt = 0). Hence, the FEM data correspond to the case of pure linear elasticity and can be directly compared to the MSM measurements. The boundary conditions used for the MSM are the same as for the FEM (stress-free boundaries).

## Local photoactivation

In the final step, we simulated the photoactivation on only the left half of the pattern. For this, we measured the spatial intensity profile and fitted a function of the form

$$I(x) = 1 - \frac{1}{1 + e^{-a(x-b)}} \tag{42}$$

to obtain the right shape given by parameters $a = 0.6497$ and $b = 13.186$. Subsequently, we modified the intensity profile such that it reaches a constant level $f$ as $x \to \infty$

$$\tilde{I}(x) = (1 - f)\left1 - \frac{1}{1 + e^{-a(x-b)}}\right + f. \tag{43}$$

The parameter $f \in [-1, 1]$ controls an active stress level on the nonactivated side and is referred to as the *degree of active coupling*. Positive and negative values for $f$ correspond to active contraction and active relaxation, respectively. The intensity profile and corresponding fit are shown in *Appendix 1—figure 1D*, while the activation profile $\tilde{I}(x)$ for different values of $f$ can be seen in *Figure 4B*. The time-dependent opto-stress tensor is modified by the spatial distribution of the intensity profile (To keep the activation profile static in the lab-frame [Eulerian frame], we incorporate the, although in many cases negligible, deformation by shifting the activation profile according to the displacement field of the previous time step such that $I(x) = \hat{I}(X + u_x)$. Here, the coordinate $X$ is fixed in the material.) by multiplication

$$\tilde{\boldsymbol{\sigma}}_{\text{opto}}(x, t) = \boldsymbol{\sigma}_{\text{opto}}(t)\tilde{I}(x) \,. \tag{44}$$

## Qualitative study of local photoactivation in a singlet

This subsection is only relevant for *Figure 3—figure supplement 5C and D*. The minimal model for fluidization that we used to characterize the local photoactivation of the singlet is depicted in *Figure 3—figure supplement 5C*. Within this approach, we model the response of the singlet by simply switching from the contractile equilibrium state (KV-model) to a Maxwell fluid with viscoelastic coupling to the substrate (coupling Stokes' elements $\gamma$ and coupling springs $Y$ in series). For simplicity, we chose to use a quasi one-dimensional such that flow and contraction are assumed to happen only along the x-direction of a cell layer of length $L$. This type of Maxwell model (*Figure 3—figure supplement 5C*, right) has been used before to study the flow dynamics of stress fibers (*Oakes et al., 2017*). Further, we allow the viscous coupling $\gamma$ to be different in the activation region. This

is assumed to artificially introduce a symmetry break between activated and nonactivated region as could be observed in experiments (*Figure 3—figure supplement 5D*, right). At this point it should be noted that, although we use the terms 'activated' and 'nonactivated,' we do not introduce active stresses but simply switch the model. We note that qualitatively similar results can be obtained by variation of the elastic modulus $E_c$ between activated and nonactivated region. Physically, this rather heuristic approach allows material flow toward the activation region. With regard to the actin intensity measurements in *Figure 3F*, we identify this as the net flow of actin. In the following, we will only derive the weak form of the quasi one-dimensional Maxwell model. The corresponding weak form for the Kelvin–Voigt model can be derived analogously. For simplicity, we further introduce the short-hand notation for the time derivative of a quantity $u$ as $\dot{u} \equiv \partial_t u$. The constitutive relation of the active Maxwell model is given by

$$\sigma - \sigma^{\mathrm{bck}} + \tau_c(\dot{\sigma} - \dot{\sigma}^{\mathrm{bck}}) = E_c \tau_c \dot{\epsilon}, \tag{45}$$

where $\tau_c = E_c/\eta_c^{\mathrm{MW}}$ is the relaxation constant for the Maxwell fluid, $\epsilon = \partial_x u(x, t)$ is the one-dimensional strain expressed in terms of the displacement field $u$, and $\sigma^{\mathrm{bck}}$ is the active background stress that is assumed to be constant ($\dot{\sigma}^{\mathrm{bck}} = 0$). Additionally, we assume stress-free boundaries $\sigma(x = (0, L), t) = 0$, which corresponds to the assumption that flow of material is sustained by creation of new actin at the ends of the cell layer. Further, we note that Stokes' friction (represented by circles) and elastic foundation (represented by springs) are in serial connection (*Figure 3—figure supplement 5C*, right). Hence, the forces acting on these elements are equal, which yields two dependent force balance equations coupled through the relation $u = u_Y + u_\gamma$.

$$h_c \frac{\partial \sigma}{\partial x}(x, t) = Y u_Y(x, t), \tag{46}$$

$$h_c \frac{\partial \sigma}{\partial x}(x, t) = \gamma \dot{u}_\gamma(x, t), \tag{47}$$

where we again use the thin-layer approximation by multiplication with the effective height of the cell layer $h_c$ and further by $u_Y$ and $u_\gamma$ denote the displacement of the Stokes' and spring element, respectively. Next, we take the derivative of *Equation (45)* with respect to $x$, which yields

$$\partial_x \sigma + \tau_c \partial_x \dot{\sigma} = \eta_c \partial_x^2 \dot{u}. \tag{48}$$

Using the time derivative of *Equations (46)* and *(47)* gives the final system of equations

$$Y u_Y + \tau_c Y \dot{u}_Y = \eta_c \partial_x^2 \dot{u} \tag{49}$$

$$\gamma \dot{u}_\gamma + \tau_c \gamma \ddot{u}_\gamma = \gamma(\dot{u} - \dot{u}_Y) + \tau_c \gamma(\ddot{u} - \ddot{u}_Y) = \eta_c \partial_x^2 \dot{u}. \tag{50}$$

In addition, the stress-free boundaries yield

$$-\sigma^{\mathrm{bck}} = \eta_{cx} \dot{u}. \tag{51}$$

Consequently, multiplying *Equations (49)* and *(50)* with test functions $w_1, w_2 \in \mathcal{D}([0, L])$ leads to

$$\int_0^L Y u_Y w_1 x + \int_0^L \tau_c Y \frac{u_Y - u_{n,Y}}{\Delta t} w_1 x$$
$$+ \int_0^L \eta_{cx} \frac{u - u_n}{\Delta t}_x w_1 x + \sigma^{\mathrm{bck}} w_1 \Big|_0^L = 0, \tag{52}$$

and

$$\int_0^L \gamma \frac{u_\gamma - u_{n,\gamma}}{\Delta t} w_2 x + \int_0^L \tau_c \gamma \frac{u_\gamma + u_{n-1,\gamma} - 2u_{n,Y}}{\Delta t^2} w_2 x$$
$$+ \int_0^L \eta_{cx} \frac{u - u_n}{\Delta t}_x w_2 x + \sigma^{\mathrm{bck}} w_2 \Big|_0^L = 0, \tag{53}$$

where we did not replace $u_\gamma$ by $u - u_Y$ for notational simplicity. For the time discretization, we use a backward Euler scheme and second derivatives with respect to time are approximated by

$$\partial_t^2 u = \frac{u + u_{n-1} - 2u_n}{\Delta t^2}, \tag{54}$$

where the indices $n$ and $n-1$ denote the two previous time steps. For the plots in *Figure 3—figure supplement 5D* (left), we calculated the strain energy according to

$$U_s = \frac{1}{2} \int_0^{L_x} Y L_y u_Y^2 \, \mathrm{dx}, \tag{55}$$

as we only consider contraction in the $x$-direction for this simplified model.

## Connection to main text figures

In *Appendix 1—figure 3* and *Appendix 1—figure 4*, we show the results as obtained by optimizing the active Kelvin–Voigt model (cf: 'Active Kelvin–Voigt model,' 'Cell–substrate coupling,' 'Parameterization,' and 'Finite element simulation') against the global photoactivation (full opto-stimulation) of doublets and singlets. The parameters corresponding to the figures are listed in *Appendix 1—table 1* and *Appendix 1—table 2*. The discrepancies between theoretical and experimental curves in *Appendix 1—figure 3C and E* and *Appendix 1—figure 4C and E* are likely explained by two points: first, TFM involves inferring forces from noisy displacement data, leading to a smoothened force field due to the introduction of a regularization parameter in the force calculations (*Schwarz and Soiné, 2015*). Second, we use a homogeneous attachment model, while in reality, cells place focal adhesions mainly in the corners and center of the pattern (*Figure 1—figure supplement 1*).

In case of local photoactivation, we used the parameterization from global photoactivation and simply introduced the spatially varying intensity profile as a function of the degree of active coupling as defined in *Equation (43)*. In *Figure 3C* (right), we show qualitative results of simulated traction force increase during local photoactivation. Here we used a value of $f = 0.3$ for doublets and $f = -0.5$ for singlets. A comparison to the case of $f = 0$ for both doublets and singlets can be found in *Figure 3—figure supplement 5A* (right) and B (solid lines).

In *Figure 4A–C*, we quantified the degree of active coupling by running several simulations for a varying degree of active coupling. To obtain these plots, we fixed all parameters as obtained by the baseline and full stimulation strain energy fits (for the doublet), and then varied the degree of active coupling $f$ as a free parameter ranging from –1 to 1 in steps of $\Delta f = 0.1$; in other words, we increased the active response on the nonactivated side in steps of 10%. For each value of $f$, the stress difference $\Delta\sigma_{xx}(x, y)$ and $\Delta\sigma_{yy}(x, y)$ between baseline and maximum strain energy was then averaged over the y-axis (*Figure 4B*). After that, the resulting x-profiles were normalized by integrating the right half of the curves and dividing that by the integral of the whole curve. This procedure allowed us to translate the family of curves (*Figure 4B*) into a relationship between the normalized stress response for $\sigma_{xx}$ and $\sigma_{yy}$ and the degree of active coupling $f$ (*Figure 4C*, solid lines). For the theoretical stress difference maps in *Figure 4A*, we used a value of $f = 0.0$ for $\Delta\sigma_{xx}$ and $f = 0.2$ for $\Delta\sigma_{yy}$. Since we are using a linear elastic constitutive relation, total stresses and hence total stress differences scale linearly with the active, that is, photoactivation stresses. Consequently, as the solid lines in *Figure 4C* are deduced by calculating the ratio of two integrals of the stress distribution along the x-direction, they should be independent of the absolute value of the active stress. We have tested this empirically and confirmed this hypothesis.

For *Figure 5F*, we repeated the same analysis as for the doublets but now for the aspect ratios of 1 to 2 and 2 to 1. Since we had no global photoactivation data, we only fitted the baseline and report the values in *Appendix 1—table 3*. Following the idea of the remark above, panel F can be reproduced with an arbitrary value of the photoactivation stress. However, to be in the right ballpark, we used the value as for the 1 to 1 doublets.

For *Figure 6F*, we repeated the same analysis as for the doublets but now for the tissue geometry (fully adherent elastic sheet). In simulations, we used a background stress similar to the measured average stresses (compare *Figure 6B*, bottom row) $\sigma_{xx}^{\mathrm{bck}} = 6.5\,\mathrm{kPa}$ and $\sigma_{yy}^{\mathrm{bck}} = 1.15\,\mathrm{kPa}$, which corresponds to a mechanical polarization of MP = 0.7. For photoactivation, we again chose the values for the doublet.

For *Figure 3—figure supplement 5D*, we qualitatively compare the time course of substrate strain energies of simulations and experiments. In simulations, we first equilibrated the cell layer using a KV model and then subsequently at $t = t_{\mathrm{act}}$ (blue vertical line) switched to an active Maxwell model.

For intermediate values of $\gamma \approx 0.3\,\mathrm{N\,s\,m^{-1}}\,\mu m^2$, with an order of magnitude as reported in the work by *Oakes et al., 2017*, the model switch leads to a transient behavior of the strain energy. Before the model switch, all elastic energy is stored in the spring of the substrate. Right after the model switch, the system starts to deform. At first, a high rate of deformation leads to a 'stiff' Stokes' element, which in turn leads to larger deformations of the substrate spring that is followed by an increase in the substrate strain energy. Over time, the deformation rate slows down and the system starts to 'flow' such that the strain energy drops below the baseline level as the substrate deformation decreases. To break the symmetry between activated and nonactivated region (in other words, to achieve a flow of material toward the activation region as observed in experiments), we ran several simulations and found that choosing a value $\gamma/2$ for the Stokes' elements on the nonactivated side qualitatively reproduces the time course and symmetry break as observed in experiments (*Figure 3—figure supplement 5D*, right). The strain energy was calculated by separately integrating the left and right half of the cell layer using *Equation (55)*. All other parameters used in this qualitative simulation are gathered in *Appendix 1—table 4*.

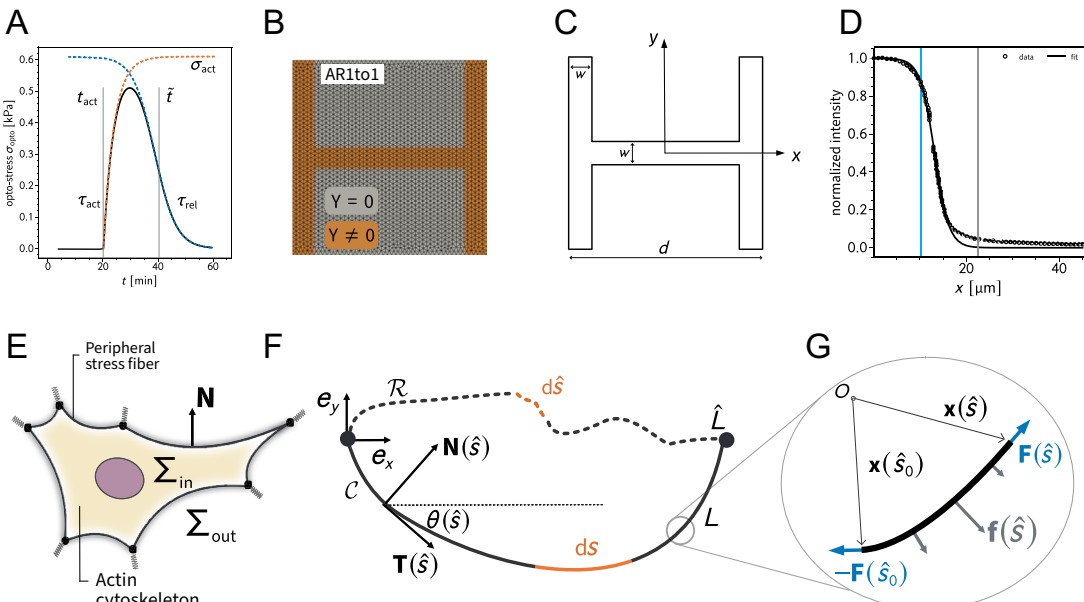

**Appendix 1—figure 1.** Overview of the 2D finite element simulation and the notation for the contour model. (**A**) shows the shape of the by optogenetic-induced time-dependent stress. (**B**) depicts the finite element mesh created with GMSH. The spring stiffness density is nonzero on the brown part of the domain. (**C**) is a schematic illustration of the relevant parameters to define the adhesion geometry. (**D**) shows the experimentally measured intensity profile of the light pulse used for photoactivation. The gray line indicates the center of the pattern (measured from left to right) while the blue line marks the inflection point of the sigmoidal fit function. (**E**) is a schematic illustration of the relevant quantities in the contour-based description of cellular adhesion. (**G**, **F**) explain the relevant mathematical quantities to describe the equilibrium shape of a fiber subject to external loads.

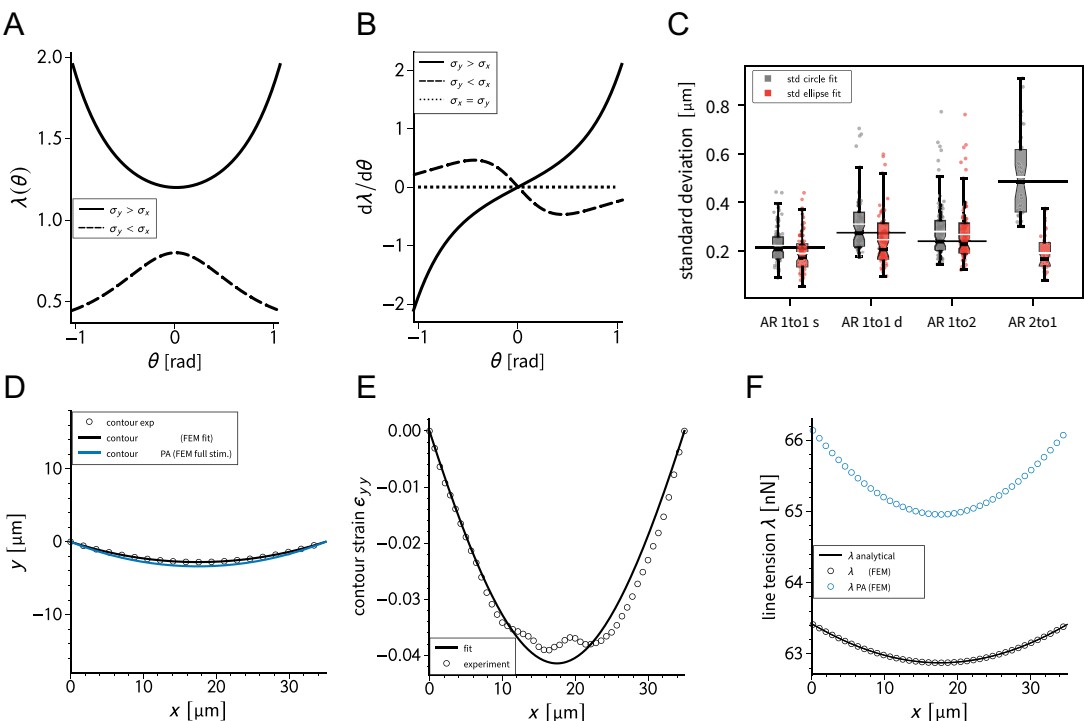

**Appendix 1—figure 2.** Calibration check for the contour model. (**A**, **B**) show the predicted line tensions and the derivative with respect to the turning angle based on the analytical solution for different values of $\sigma_x$ and $\sigma_y$. (**C**) compares the circle and ellipse fit of the contour of the cells for different pattern aspect ratios. (**D**) shows a generic cell contour for the doublet before and during photo-activation. The experimentally contour strain in $y$-direction with the respective fit from simulations and the corresponding line tensions are shown in (**E**) and (**F**), respectively.

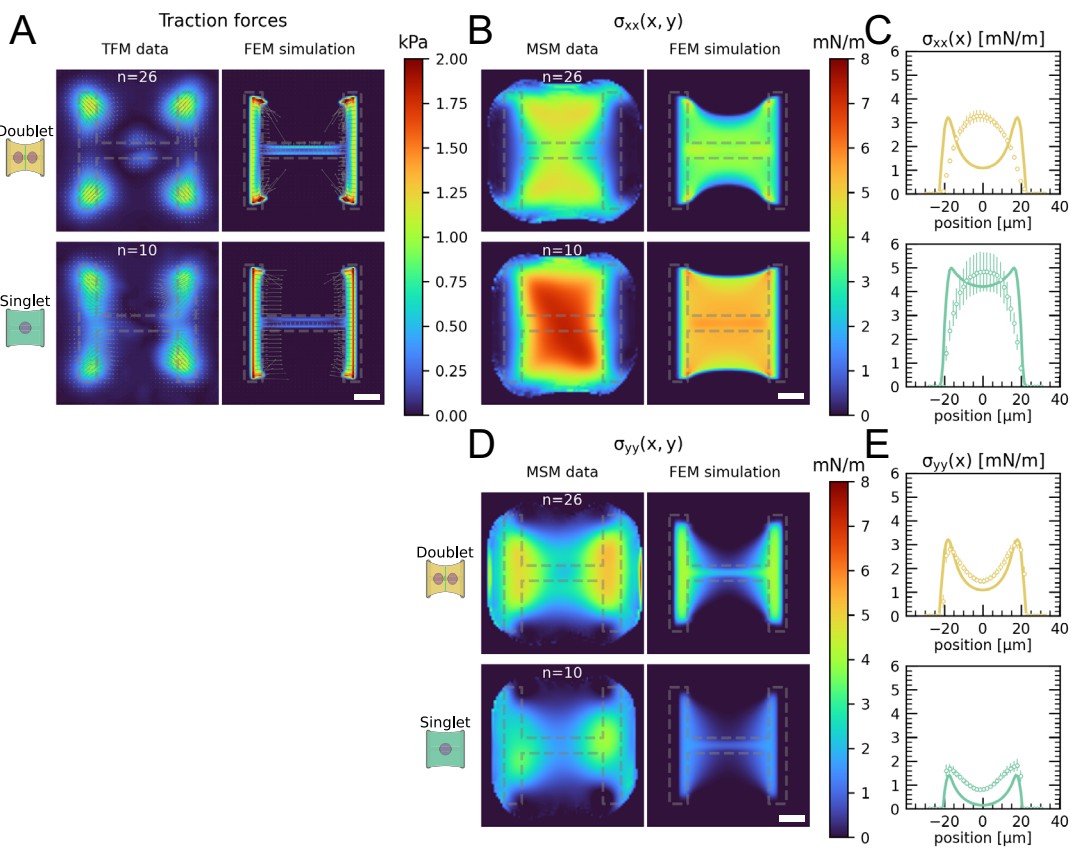

**Appendix 1—figure 3.** Comparison of traction forces and stress distribution between experiment and finite element simulation. (**A**) Average traction stress and force maps of cell doublets (top) and singlets (bottom) on H-patterns on the left and corresponding traction stress and force maps from the FEM simulation. (**B**, **D**) Average cell stress maps of cell doublets (top) and singlets (bottom) on H-patterns on the left and corresponding cell stress maps from the FEM simulation. (**C**, **E**) Average over the y-axis of the maps in (**B**) and (**D**). Data is shown as circles with the mean ± SEM, the solid line corresponds to the FEM simulations. All scale bars are $10\,\mu$m long.

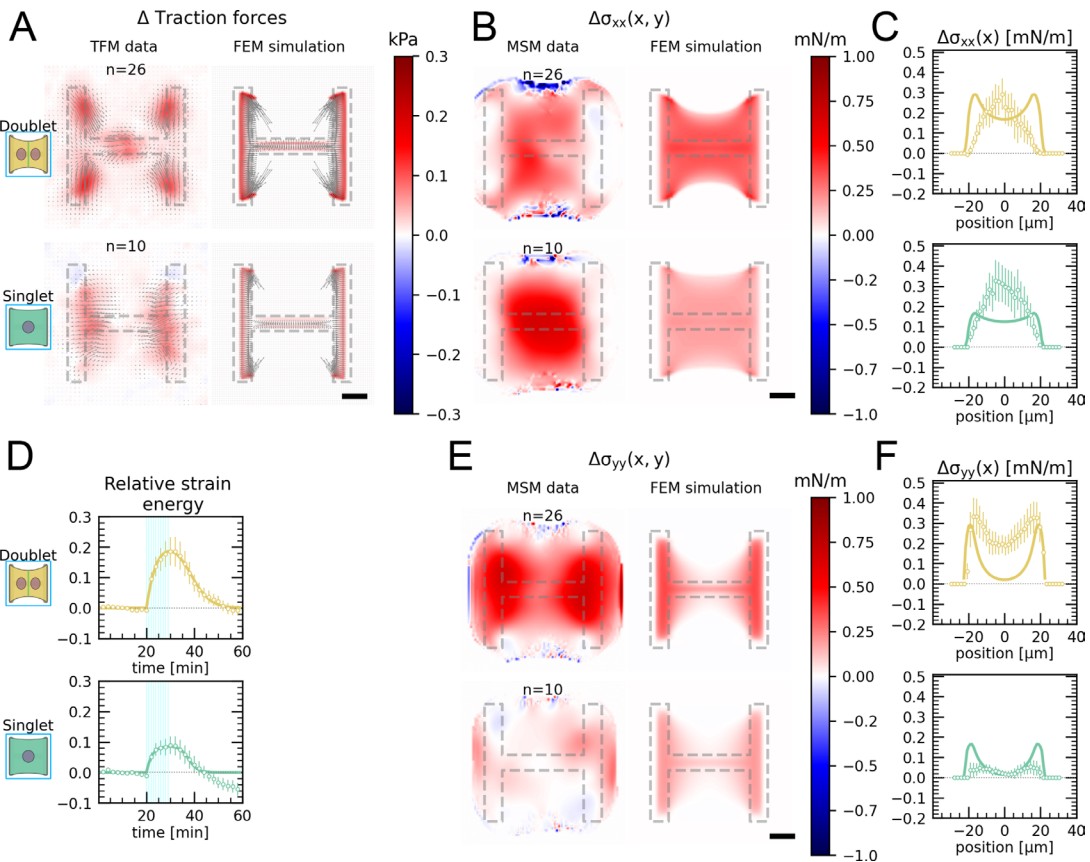

**Appendix 1—figure 4.** Comparison of traction forces stress distribution and strain energy between experiment and finite element simulation for global photoactivation. (**A**) Average traction stress and force map difference before and after photoactivation of cell doublets (top) and singlets (bottom) on H-patterns on the left and corresponding traction stress and force maps from the FEM simulation. (**B, E**) Average cell stress map difference before and after photoactivation of cell doublets (top) and singlets (bottom) on H-patterns on the left and corresponding cell stress maps from the FEM simulation. (**C, F**) Average over the y-axis of the maps in (**B**) and (**D**). Data is shown as circles with the mean ± SEM, the solid line corresponds to the FEM simulations. (**D**) Relative strain energies of doublets (top) and singlets (bottom) with global photoactivation. One frame per minute was acquired for 60 min, and cells were photoactivated with one pulse per minute for 10 min between minute 20 and minute 30. Strain energy curves were normalized by first subtracting the individual baseline energies (average of the first 20 min) and then dividing by the average baseline energy of all cell doublets/singlets in the corresponding datasets. Data is shown as circles with the mean ± SEM, and the result of an FEM simulation is shown as a solid line. All scale bars are $10\,\mu$m long.

## Contour model

The observed invaginated arcs in strongly adherent cells (*Figures 1A and 2B*) can be geometrically explained by the interplay between a surface tension $\sigma$ associated with the contractile cortex and the resisting line tension $\lambda$ in the strong peripheral actin bundle. In case of a homogeneous cortex, one may assume the surface tension to be isotropic, which yields a Laplace law predicting a constant radius of curvature $R = \lambda/\sigma$ (*Bischofs et al., 2008*; *Barziv et al., 1999*; *Vianay et al., 2010*). Moreover, the observed dependence of the curvature of the arc on the spanning distance $d$ of the two endpoints can be explained by assuming an elastic contribution to the line tension (*Bischofs et al., 2008*). This modification of the simple tension model (STM) is known as the tension elasticity model (TEM) and yields a relationship $\lambda(d)$, which in turn leads to an increasing $R$-$d$ relationship. However, in some cases the assumption of a homogeneous isotropic cortex fails in the presence of strongly embedded internal stress fibers. In this scenario, the isotropic surface tension is modified by a directional component aligned with the direction of the internal stress fibers. This so-called ATM predicts elliptical arcs and a position-dependent line tension in the fiber (*Pomp et al., 2018*). A comprehensive summary of the different types of existing contour models can be found in *Giomi, 2019*.

## Anisotropic surface tension

Like all contour models, the ATM is based on a very general force balance equation for a slender fiber, which we will motivate very briefly. The fiber is assumed to be resistant to tension only such that bending and shearing are neglected. Further, we assume the fiber to start and end at discrete fixed points, which resemble the focal adhesions. Each fiber has a reference shape (unstrained, stress free) and a current configuration (strained). All quantities associated with the reference shape are denoted by a ^-symbol (*Appendix 1—figure 1F*).

The resulting surface tension acting on the edge bundle is given by the difference of the interior and exterior stress tensors (*Appendix 1—figure 1E*). Since the micropattern in all our experiments has two symmetry axes, we assume an anisotropic surface tension tensor of the form

$$\Sigma_{\text{out}} - \Sigma_{\text{in}} = \begin{pmatrix} \sigma_x & 0 \\ 0 & \sigma_y \end{pmatrix} . \tag{56}$$

By introducing a Frenet–Serret frame as a local basis to the current configuration of the fiber

$$\frac{d\mathbf{x}}{ds} = \mathbf{T} \tag{57}$$

$$\frac{d\mathbf{T}}{ds} = \kappa \mathbf{N}, \tag{58}$$

where $s$ denotes the arc-length parameter along the current state, $x$ the shape of the current state, and $\kappa$ the local curvature (*Appendix 1—figure 1F*), one can derive the force balance equation by considering an infinitesimal line element in the current configuration as illustrated in *Appendix 1—figure 1G*. For such a line element, the force balance reads

$$\frac{d}{ds}\mathbf{F}(s) + \begin{pmatrix} \sigma_x & 0 \\ 0 & \sigma_y \end{pmatrix} \mathbf{N}(s) = 0, \tag{59}$$

where $\mathbf{F}(s) = \lambda(s)\mathbf{T}(s)$ always points tangential to the fiber with line tension $\lambda(s)$. Finally, it can be shown that *Equation (59)* leads to the equation of an ellipse

$$\frac{y^2}{C\sigma_y} + \frac{x^2}{C\sigma_x} = 1 , \tag{60}$$

with semi-axes given by $a = \sqrt{C\sigma_x}$ and $b = \sqrt{C\sigma_y}$. In the isotropic case, for which $\sigma_x = \sigma_y$, the ellipse attains circular shape consistent with the results of the STM and TEM.

The line tension is now a complicated function of the turning angle $\theta(s)$ given by

$$\lambda(\theta) = \sigma_x \sqrt{\sigma_y C} \sqrt{\frac{1 + \tan^2 \theta}{1 + \frac{\sigma_x}{\sigma_y} \tan^2 \theta}} . \tag{61}$$

By taking derivative of this expression with respect to the turning angle $\theta$, one can show that the line tension has an extremum at $\theta = \theta_0 = 0$ given by

$$\lim_{\theta \to 0} \lambda(\theta) = \sigma_x \sqrt{\sigma_y C} . \tag{62}$$

Depending on the ratio, this extremum is either a maximum for $\sigma_x/\sigma_y > 1$ or a minimum for $\sigma_x/\sigma_y < 1$. In case of $\sigma_x = \sigma_y$, we obtain a constant line tension independent of the turning angle. Plots of the line tension and its derivative are shown in *Appendix 1—figure 2A and B*.

## Shape analysis

Analyzing the cell shape is equivalent to quantifying the minimal number of key parameters like line and surface tension based on the shape of the free spanning fiber. Our goal was to apply the ATM to the TFM and fiber tracking data (for fiber tracking data, we refer to the 'Materials and methods' section).

By means of our analysis, we assume that all traction contribution stems from the combined action of the free spanning arc and the vertical 'adherent' fiber of length $L$ and add up at the intersection

$$\mathbf{F}_{\mathrm{s}} = F_{\mathrm{a}}\mathbf{e}_y + \sigma_x \frac{L}{2}\mathbf{e}_x + \lambda\mathbf{T}(\theta_{\mathrm{fa}}), \tag{63}$$

where $\mathbf{F}_s$ is the force measured in the substrate, $\theta_{\mathrm{fa}}$ denotes the tangent angle at the focal adhesion, and the second term is a possible contribution of the surface tension that only has an $x$-component due to the fact that the adherent fiber is straight and aligned in y-direction (*Figure 2A–C*). It should be noted that $\sigma_y$ does not directly contribute to the traction forces, but only indirectly through the inward pull of the arc and thus changing the line tension lambda. Splitting up *Equation (63)* into the respective $x$-and $y$-components yields a system of two equations in the unknowns $F_{\mathrm{a}}$ and $\lambda$. The force $\mathbf{F}_{\mathrm{s}}$ was obtained by dividing the traction map into four quadrants and calculating the sum for each quadrant. A similar procedure was presented in *Labouesse et al., 2015*. The contribution of the $x$-component of the surface tension $\sigma_{\mathrm{x}}$ along the vertical fiber was estimated on TFM data as well as by summing up the x-traction stresses in a window around the center of the vertical fiber. For the two unknowns, we have

$$\lambda = \frac{1}{T_x}\left(F_{\mathrm{s},x} - \sigma_x \frac{L}{2}\right) \tag{64}$$

$$F_{\mathrm{a}} = F_{\mathrm{s},y} - \frac{T_y}{T_x}\left(F_{\mathrm{s},x} - \sigma_x \frac{L}{2}\right), \tag{65}$$

such that $\lambda$ and $F_{\mathrm{a}}$ can be calculated in terms of the tangent angle of the free spanning fiber at the focal adhesion.

## Ellipse shape fitting

It turned out, that fitting ellipses directly to 'short' arcs is very unstable and highly depends on the initialization of the fit parameters. This is because one can find a wide range of ellipses that fit equally well. Due to large data sets of 10–40 cells per condition, where each cell data set consists of 60 time frames, it was not feasible to fit ellipses by hand. Therefore, we decided to use a very stable and fast circle fitting algorithm to obtain an estimate for the tangent vector at the adhesion point (Although it is also possible to obtain the tangent vector directly from the fiber tracking data, we found through trial and error that this method is prone to large fluctuations.). For the circle fitting, we exploited a *Hyper least squares* algorithm presented in *Kanatani and Rangarajan, 2011* based on algebraic distance minimization. The already determined parameters from TFM data and circle fitting are $\sigma_x, \theta_{\mathrm{fa}}, \mathbf{T}(\theta_{\mathrm{fa}}), \lambda(\theta_{\mathrm{fa}})$. The remaining unknowns are the y-component of the surface tension tensor $\sigma_y$ as well as the center of the ellipse $\mathbf{x}_{\mathrm{c}}$ (We used the centers of the circles [from circle fitting] as initial guesses for the ellipse fitting.). Using *Equation (61)* evaluated at $\theta_{\mathrm{fa}}$, this yields

$$a = \frac{\lambda(\theta_{\mathrm{fa}})}{\sqrt{\sigma_x\sigma_y}}\sqrt{\frac{1 + \frac{\sigma_x}{\sigma_y}\tan^2(\theta_{\mathrm{fa}})}{1 + \tan^2(\theta_{\mathrm{fa}})}} \tag{66}$$

$$b = \frac{\lambda(\theta_{\mathrm{fa}})}{\sigma_x}\sqrt{\frac{1 + \frac{\sigma_x}{\sigma_y}\tan^2(\theta_{\mathrm{fa}})}{1 + \tan^2(\theta_{\mathrm{fa}})}} \tag{67}$$

such that the shape of the ellipse purely depends on $\sigma_y$. The fit was carried out by minimizing the squared distance of all tracking points along the fiber to the ellipse. The distance of those points to the ellipse was obtained by an elegant way to calculate the minimal distance of a point to the ellipse (https://blog.chatfield.io/simple-method-for-distance-to-ellipse/). *Appendix 1—figure 2C* compares the standard deviations for the two fits for all conditions. In all cases, the ellipse fit yield a smaller standard deviation, although the differences vary for the different aspect ratios. The results of this analysis are summarized in *Figure 2C–E*.

## Contour strain FEM method

In order to study the effect of photoactivation on the contour and quantify the degree of active coupling purely based on the shape of the contour, we developed a discretized FEM version of the force balance equation (*Equation 59*). In this context, we reformulate *Equation (59)* as a function of

the reference arc length parameter $\hat{s}$ (*Appendix 1—figure 1F*) in the reference state. The relationship between the two arc length parameters is given by the stretch

$$\nu(\hat{s}) := \left|\frac{\partial \mathbf{x}}{\partial \hat{s}}\right| = \frac{ds}{d\hat{s}} = \sqrt{(\partial_{\hat{s}} x)^2 + (\partial_{\hat{s}} y)^2}. \tag{68}$$

This allows us to express the equation of mechanical equilibrium as

$$\frac{d}{d\hat{s}}\left(\lambda(\hat{s}) \frac{1}{\nu(\hat{s})} \frac{d\mathbf{x}}{d\hat{s}}\right) + \begin{pmatrix} \sigma_x & 0 \\ 0 & \sigma_y \end{pmatrix}\left(\frac{d\mathbf{x}}{d\hat{s}}\right)_{\perp} = 0, \tag{69}$$

where $\left(d\mathbf{x}/d\hat{s}\right)_{\perp} = (dy/d\hat{s}, -dx/d\hat{s}) = \nu(\hat{s})\mathbf{N}(s(\hat{s}))$. This coupled system of equations can be solved by means of a finite element implementation with mixed elements on a one-dimensional mesh. Let $w_1, w_2 \in D([0, d])$ be two test functions over the interval $[0, d]$ representing the spanning distance of the unstretched straight fiber. Following the standard procedure by multiplying *Equation (69)* with the test functions (one test function for each equation) and integrating it over the simulation domain yields

$$-\int_0^d \lambda(\hat{s}) \frac{1}{\nu(\hat{s})} \frac{dx}{d\hat{s}} \frac{dw_1}{d\hat{s}} \, d\hat{s} + \int_0^d \sigma_x \frac{1}{\nu(\hat{s})} \frac{dy}{d\hat{s}} w_1 \, d\hat{s} = 0 \tag{70}$$

$$-\int_0^d \lambda(\hat{s}) \frac{1}{\nu(\hat{s})} \frac{dy}{d\hat{s}} \frac{dw_2}{d\hat{s}} \, d\hat{s} - \int_0^d \sigma_y \frac{1}{\nu(\hat{s})} \frac{dx}{d\hat{s}} w_2 \, d\hat{s} = 0 . \tag{71}$$

Here we used partial integration

$$\int_0^d \frac{d}{d\hat{s}}(.)w_i \, d\hat{s} = (.)w_i\big|_0^d - \int_0^d (.)\frac{dw_i}{d\hat{s}} \, d\hat{s} , \tag{72}$$

and that by construction $w_i = 0$ on the boundary. Further, we impose Dirichlet boundary conditions $x(0) = 0$, $x(d) = d$, $y(0) = y(d) = 0$ such that the endpoints of the fiber are fixed.

## Modeling procedure for the contour finite element simulation

The modeling procedure for the contour simulation is structurally very similar to the 2D version explained above. The aim was to quantify the active coupling between activated and nonactivated part of the cell doublet. The results of the contour analysis allowed us to obtain an average ellipse (corresponding to an average cell shape) by averaging the results for $a, b, \sigma_x, \sigma_y$. Based on actin images, the spanning distance of the fiber was estimated to a value of $d = 35 \ \mu$m. An average elliptical contour was created by fixing $\sigma_x$ and $\sigma_y$ as well as the semi-axis $a$. From those values, we then computed $b = a\sqrt{\sigma_y/\sigma_x}$. This was necessary since we averaged all those quantities independently of each other such that the averages of the single quantities not necessarily describe an elliptical arc. In the spirit of the TEM (*Bischofs et al., 2008*; *Bischofs et al., 2009*) and inspired by the work of *Labouesse et al., 2015*, we split the line tension into an active and elastic contribution where the first accounts for the elastic properties of the cross-linking proteins (such as $\alpha$-actinin) within the actin bundle and the latter is an active contribution from myosin II motors such that

$$\lambda = \lambda_{\mathrm{el}} + \lambda_{\mathrm{act}} . \tag{73}$$

We further assumed a linear constitutive relationship between stress and strain for the elastic component

$$\lambda_{\mathrm{el}} = \mathrm{EA}\epsilon = \mathrm{EA}(\nu(\hat{s}) - 1) , \tag{74}$$

which is directly connected to the stretch as defined in *Equation (68)*. The distinction between elastic and active force contribution is not necessary for the main conclusions of this work, but was added for consistency with the work by *Bischofs et al., 2008*; *Bischofs et al., 2009* and *Labouesse et al., 2015*. The rest length of the fiber is set to the spanning distance $\hat{L} = d$. Here, EA denotes the one-dimensional modulus of the fiber as a product of Young's modulus $E$ and the cross-sectional area $A$. This value is typically around $\mathrm{EA} = 50 \ \mathrm{nN} - 350 \ \mathrm{nN}$ (*Guthardt Torres et al., 2012*; *Labouesse et al., 2015*; *Deguchi et al., 2006*). By means of our contour simulation, we set this value to EA =300 nN. All other fixed values for this simulation can be found in *Appendix 1—table 5*. In a first step, we

minimized the simulated contour against the average contour from the contour analysis treating $\lambda_{\text{act}}$ as a free parameter (*Appendix 1—figure 2D*). Subsequently, we introduced full optogenetic stimulation by defining

$$\sigma_i^{\text{PA,max}} = \sigma_i + \sigma_i \cdot \text{RSI}_i^{\text{max}} \, , \tag{75}$$

where $\sigma_i^{\text{PA,max}}$ denotes the respective surface tension component at maximum strain energy, $\text{RSI}_i^{max}$ is the maximal relative surface tension increase and $i = x, y$. Then, we optimized the values $\text{RSI}_x^{\text{max}}, \text{RSI}_y^{\text{max}}$ to fit the measured contour strain to the one computed with the contour FEM at maximum strain energy by additionally making sure that the values for the RSI do not exceed the from statistics experimentally obtained bounds for these values. For this, we exploited a sequential least-squares programming algorithm (SLSQP) (*Kraft, 1988*) implemented in SciPy (*Virtanen et al., 2020*), which, in contrast to the simplex algorithm, allows constrained minimization. *Figure 4D* illustrates the contour strain measurement (shown for image data, but is performed in the same way in the simulation). We measure the vertical inter-stress fiber distance after ($L_{\text{PA}}^{\text{max}}$) and before ($L_{\text{bck}}$) photoactivation along the $x$-axis for each tracking point (circles). This procedure defines a contour strain, which may be defined as

$$\epsilon_{yy} = \frac{L_{\text{PA}}^{\text{max}}}{L_{\text{bck}}} - 1 \, . \tag{76}$$

The negative values for the contour strain (*Figure 4E*, *Appendix 1—figure 2E*) indicate that the free arcs move toward the cell interior during photoactivation. The result of this optimization is depicted in *Appendix 1—figure 2E*.

Finally and analogously to the two-dimensional case (*Equation 44*), local photoactivation was introduced by

$$\sigma_i^{\text{PA,max}}(\hat{s}) = \sigma_i + \sigma_i \cdot \text{RSI}_i^{\text{max}} \cdot \tilde{I}(\hat{s}) \, , \tag{77}$$

For different values for the degree of active coupling $f$, we simulated the contour strain leading to the family of curves as depicted in *Figure 4E*. The response of the nonactivated side as a function of the degree of active coupling was then obtained by the integral of the right half of the curve divided by the integral of the whole curve (*Figure 4F*).

### Connection to main text figures

In *Figure 2B–E*, we show the results as obtained by the shape analysis as explained in subsections 'Shape analysis' and 'Ellipse shape fitting.' In *Figure 4D–F*, we show the results as obtained in the context of contour strain simulations. Thereby, *Figure 4D* (right panel) exemplarily shows a contour simulated with a spatially varying surface tension according to *Equation (77)* and an intensity profile of the light pulse as depicted in *Appendix 1—figure 1D*. *Figure 4E and F* were obtained by following the procedure explained in subsections 'Contour strain FEM method' and 'Modeling procedure' for the contour finite element simulation.

**Appendix 1—table 1.** Fixed parameters for the two-dimensional finite element simulation.

| Fixed parameter | Value |
|---|---|
| Substrate | |
| Young's modulus of the substrate, $E_s$ | 20 kPa |
| Poisson's ratio of the substrate, $\nu_s$ | 0.5 |
| Thickness of the substrate, $h_s$ | 50 $\mu$m |
| Cell | |
| Young's modulus of cell, $E_c$ | 10 kPa |
| Viscosity of the cell, $\eta_c$ | 100 kPa s |

*Appendix 1—table 1 Continued on next page*

*Appendix 1—table 1 Continued*

| Fixed parameter | Value |
| --- | --- |
| Effective thickness of the cell, $h_c$ | 1 $\mu$m |
| Poisson's ratio of the cell, $\nu_c$ | 0.5 |
| Length of the cell, $L_c$ | 50 $\mu$m |

**Appendix 1—table 2.** Fit parameters as obtained by the two-dimensional finite element simulation.

| Fit parameter | Singlet | Doublet |
| --- | --- | --- |
| Baseline | | |
| Background stress component, $\sigma_{xx}^{\text{bck}}$ | 6.59 kPa | 5.73 kPa |
| Background stress component, $\sigma_{yy}^{\text{bck}}$ | 2.78 kPa | 5.73 kPa |
| Full opto-stimulation | | |
| Active stress, $\sigma_{\text{act}}$ | 0.287 kPa | 0.618 kPa |
| Activation time scale, $\tau_{\text{act}}$ | 133 s | 227 s |
| Relaxation time scale, $\tau_{\text{rel}}$ | 113 s | 236 s |
| Centroid, $\tilde{t}$ | 1057 s | 1117 s |

**Appendix 1—table 3.** Fit parameters for baseline fit of AR 1 to 2 and AR 2 to 1 doublets. The background stress was obtained from a fit of the model to the strain energy baseline. Based on the values for the mechanical polarization reported in *Figure 5G*, we used MP = 0.46 for AR 1 to 2 and MP = $-0.61$ for AR 2 to 1. Together with the fitted values for $\sigma_{xx}^{\text{bck}}$ we deduced $\sigma_{yy}^{\text{bck}}$ from *Equation (17)*.

| Fit parameter | AR 1 to 2 | AR 2 to 1 |
| --- | --- | --- |
| Baseline | | |
| Background stress component, $\sigma_{xx}^{\text{bck}}$ | 6.4 kPa | 2.2 kPa |
| Background stress component, $\sigma_{yy}^{\text{bck}}$ | 2.4 kPa | 8.9 kPa |

**Appendix 1—table 4.** Fixed parameters used in the qualitative study of fluidization. The order of magnitude for $\eta_c^{\text{MW}}$ was set in accordance with the values reported in *Oakes et al., 2017*.

| Fixed parameter | Value |
| --- | --- |
| Young's modulus of the substrate, $E_s$ | 20 kPa |
| Poisson's ration of the substrate, $\nu_s$ | 0.5 |
| Effective substrate thickness, $h_s$ | 50 $\mu$m |
| Lateral cell size, $L_c$ | 45 $\mu$m |
| Young's modulus of the cell, $E_c$ | 10 kPa |
| Viscosity of the cell, $\eta_c^{\text{MW}}$ | 10 MPa s |
| Effective thickness of the cell, $h_c$ | 1 $\mu$m |
| Background stress, $\sigma^{\text{bck}}$ | 5 kPa |
| Spring stiffness density, $Y$ | 1.26 mNm$^{-1}$ $\mu$m$^{-1}$ |

**Appendix 1—table 5.** Fixed and optimized parameters for the contour shape analysis by means of the contour finite element simulation.

| Parameter | Value |
| --- | --- |
| Fixed | |
| Surface tension component, $\sigma_x$ | $0.92\,\mathrm{nN}\,\mu\mathrm{m}^{-1}$ |
| Surface tension component, $\sigma_y$ | $1.12\,\mathrm{nN}\,\mu\mathrm{m}^{-1}$ |
| Semi-axis, $a$ | $61.94\,\mu\mathrm{m}$ |
| Semi-axis, $b$ | $68.34\,\mu\mathrm{m}$ |
| One-dimensional elastic modulus, $EA$ | $300\,\mathrm{nN}$ |
| Contour fit | |
| Active line tension, $\lambda_{\mathrm{act}}$ | $58.1\,\mathrm{nN}$ |
| Strain fit | |
| Relative surface tension increase, $\mathrm{RSI}_x^{\max}$ | $0.11\,\mathrm{nN}\,\mu\mathrm{m}^{-1}$ |
| Relative surface tension increase, $\mathrm{RSI}_y^{\max}$ | $0.24\,\mathrm{nN}\,\mu\mathrm{m}^{-1}$ |

