## [Editor Report]

Using surface micropatterning, optical activation, and theoretical analysis, the authors provide compelling evidence that adjacent cells actively propagate mechanical stress in epithelial tissues. The response of the receiver cell is active and enhanced when the principal stress direction is perpendicular to the orientation of actin fibers. This work is important and a must read for everybody wanting to understand tissue mechanics.

---

## [Decision Letter]

**Decision letter after peer review:**

Thank you for submitting your article "Force propagation between epithelial cells depends on active coupling and mechano-structural polarization" for consideration by *eLife*. Your article has been reviewed by 2 peer reviewers, one of whom is a member of our Board of Reviewing Editors, and the evaluation has been overseen by Jonathan Cooper as the Senior Editor. The reviewers have opted to remain anonymous.

Essential revisions:

1) The link between the supplementary information and the main text needs to be improved as the connection between some SM sections and the corresponding figures or claims in the main text is sometimes unclear. In many figures, it remains unclear which theoretical description was used, what is the underlying hypothesis that was made, or the parameter values. This limits the reproducibility of results. See the report of Reviewer #2 for details.

2) Clarify that MSM relays on some hypothesis (linear elasticity) to infer stresses. See the report of Reviewer #2 for details.

*Reviewer #2 (Recommendations for the authors):*

There are a few concerns. The linkage between the supplementary material and the main text can be improved (see point 1). To reproduce the theoretical panels, additional details are required (see point 1). Comparison of MSM stresses and theory (see point 2).

1. The linkage between the supplementary information and the main text can be improved. The connection between some SM sections and the corresponding figures or claims in the main text is unclear in many cases. In many figures, it is unclear which was the theoretical description used, the underlying hypothesis made, or the values of parameters, which limits the reproducibility of some results.

1.1. In Figure 2A, the authors provide a schematic of the contour plot. According to Equation 48 in SM, the total traction force F_s is a combination of the tension of the free edge, the tension (σ_x) of the vertical fiber, and a force F_ad (in the schematic is called F_a, is the same?). In the SM, F_ad is undefined. No details on how was σ_x estimated can be found. Is there a contribution of σ_y in the force balance 48? In Section 2.1, the authors define σ_x and σ_y as anisotropic surface tension. Are these related to the stresses obtained from MSM? In Section 2.2.1, can the authors explain in more detail how was σ_x estimated from TFM data and explain how the center of the ellipse was set?

1.2. In Figure 3C, can the authors explain how were the two theoretical panels obtained and what parameter values used? In Section 1.4 of the SM, there is confusion between u, Y, and us and Ys. Can the authors clarify the physical definition of each of them? It is unclear to me that the tractions can be linearly related to the displacement of the substrate without approximations. In A. F. Mertz (2011), the authors write a similar expression by ignoring non-local elastic effects. Can the authors comment on this point? In Equations 21, the authors assume that the displacement of the cell and the substrate are equal but no justification for this approximation is provided. The derivation of the effective substrate height Equation 20 was not found. It is unclear why the substrate height depends on the cell layer size, can the authors comment on this point?

1.3. In Figure 4, a scale of the active coupling f for panels B and E is missing. Can the authors add them? Can the authors provide a physical explanation for why the S_xx on the left half depends on f, and S_yy on the left half does not? According to Equation (15), it seems that the Σ_opto is isotropic. It is unclear how do the author obtain a different active coupling for S_xx in panel C top and for S_yy in panel C bottom. This comment applies also to Figure 5F and 6F. Can the authors provide more detail on how f was estimated? Besides, I was not able to understand how were panels 4E and 4F obtained. Can the authors explain this in SM?

1.4. Similar comments apply to Supplementary figures, S4A, B and D, and S5C. Besides in S4, the authors use a Maxwell model instead of the Kelvin-Voigt that is used in the main text, however, no details on the SM could be found for the Maxwell model (values of the parameters, theoretical description, numerical solvers).

Several of the above concerns arise because the linkage between the SM and the main text can be improved. I would recommend making sections in the SM that focus in more detail on how each theoretical panel was obtained with enough detail so that these panels can be reproduced. Then these sections can be referenced in the main text or captions for the corresponding analyses. Note also that the order of SM sections does not correspond to the order of appearance in the main text.

2. The authors use MSM data to compare with their theory. MSM is a method that infers stresses from traction force data by assuming that the material behaves as a linear elastic material with a fixed elastic modulus and Poisson ratio. In several parts of the main text, it should be clarified that MSM relays on some hypothesis (linear elasticity) to infer stresses (page 7 below Section "The presence of an intercellular…"). The MSM stress data are compared to theoretical predictions of a Kelvin-Voigt viscoelastic model with anisotropic active stresses. There are simple configurations where both models have different stress distributions for the same traction force distribution and the same boundary conditions. It is unclear to me that the two stresses can be expected to be comparable in the geometry of the experiments (2d square). Can the authors comment on this point?

[Editors’ note: further revisions were suggested prior to acceptance, as described below.]

Thank you for resubmitting your work entitled "Force propagation between epithelial cells depends on active coupling and mechano-structural polarization" for further consideration by *eLife*. Your revised article has been evaluated by Jonathan Cooper (Senior Editor) and a Reviewing Editor.

The manuscript has been improved but there are some remaining issues that need to be addressed, as outlined below:

Unfortunately, we had not previously managed to secure a review by an experimentalist on the original manuscript. We have now obtained a report commenting specifically on the experimental part and we invite you to consider the recommendations given by this referee (Reviewer #3). The reviewer requests additional control experiments that seem straightforward and will significantly support the conclusions.

*Reviewer #3 (Recommendations for the authors):*

In this paper, Ruppel and colleagues explore the transmission of forces from cell to cell using a minimalistic cellular system consisting of two epithelial cells seeded on a H micropatterned shape. With this strategy, they unravel a longstanding question of the active or passive nature of force transmission in epithelia. By comparing doublets of cells with single cells and using optogenetics to stimulate contraction, the authors elegantly demonstrate that cells actively propagate forces from neighboring cells. They do precise measurements with traction force microscopy and monolayer stress microscopy. Finally, they complement their analysis with computational modeling and a mini epithelial system constituted by 10-20 cells.

This is an elegant piece of research, addressing the transmission of forces in epithelial cells with a minimalistic system. This system has the clear advantage of being very controlled and allowing precise and repeatable measurements. The demonstration that a cell (the receiver in the optogenetic experiments) can actively receive and produce forces in response to its neighbor (the sender in the optogenetic experiments), is in our opinion the most important finding of the paper and justifies per se the publication.

While this is a significant demonstration, the simplicity of the system might overlook the complexity of force transmission on a real epithelial tissue in a living organism that has different mechanical properties.

Because of our expertise and the request of the editors, we will focus our analysis only on the experimental part. This complements the reviews of the other reviewers that focused more on the computational modeling part.

We think that this paper is compelling and provides significant advances in the field of epithelial biology. For this reason, we consider it appropriate for *eLife* and important for the scientific community in the field.

However, we feel that some more controls and experiments are needed to fully support the claims of the article. We will be glad to recommend the article for publication if the authors can answer our concerns 1, 2, and 3 by experiments, and 4 and 5 by discussion.

The optogenetic experiment (activation of the left cell and the observed contraction of the right cell) is central to this article, and the accidental activation of the right cells by light should be more carefully excluded. Indeed, various occurrences can lead to the accidental activation of the right cell, such as light scattering or protrusions. The demonstration that the authors provide in Figure 3 suppl 2, doesn't fully exclude accidental activation of the right cell.

1. Protrusions. Cells could emit protrusions that go from the right cell underneath the left cell and thereby being activated when the left cell is stimulated. Especially in the context of Figure 5 where the degree of active coupling is not only correlated with structural/mechanical polarization, but also the length of the shared border between cells (larger area where protrusions might be accidentally activated in 2 to1 patterns vs 1to2 patterns). This can be excluded by, for instance, using a genetically-encoded membrane marker expressed only in one of the two cells.

2. Accidental light on the right cell. This could be easily confuted by having doublets with only the right cell expressing the optogenetic RhoA tool and illuminating the left cell that doesn't have the tool. In this case, no contraction should be observed in the right cell.

3. If we assume a similar expression of optogenetic tool in each cell, the density of optogenetic receptors per area in doublets should be twice the density of singlets, leading to stronger activation of RhoA. Could this effect e.g. explain a larger increase of strain in doublets vs singlets? Or would results remain the same if light dose was halved in doublets?

4. While not clear what the molecular mechanism of transmission is (and out of the scope of this study), inhibition of e-cadherins binding (or similar) could be a way to disrupt force propagation into neighboring cells. If no force response is measured in the neighboring cell, this would also help the claim that the neighboring cell does not receive sufficient light to activate the optogenetic system. Also, such a result would demonstrate that cells require direct contact and no soluble factor is involved in the activation of the receiver. Have the authors done any attempt to perturb the signal propagation using drugs? If not, can the authors discuss this important aspect?

5. On the formation of doublets: In the text, it is explained that: "most doublets have started as single cells and divided on the pattern to form a doublet."

Was this difference controlled for? How did the rest of the doublets form? Were they seeded as doublets initially? Is there a factor (e.g. cell size) that biases which cells seed as doublets vs cells that seed alone?

For the doublets formed after division: Is there any bias comparing cells after mitosis (in doublet) vs cells before mitosis (singlet), could cell cycle state explain e.g. the increased abundance of stress fibers in singlets? Thoughts about synchronizing cell cycle state?

From the text, some questions remain regarding the method of selecting stimulation regions in optogenetic experiments.

Were the regions selected automatically or manually?

If the stimulation area was aligned using fluorescent Fibrinogen Alexa488, does this not activate the CRY2 system? If aligned on actin marker, how were stimulations kept of consistent area between repeats?

How was stimulated, a single plane in confocal?

---

## [Author Response]

Essential revisions:1) The link between the supplementary information and the main text needs to be improved as the connection between some SM sections and the corresponding figures or claims in the main text is sometimes unclear. In many figures, it remains unclear which theoretical description was used, what is the underlying hypothesis that was made, or the parameter values. This limits the reproducibility of results. See the report of Reviewer #2 for details.2) Clarify that MSM relays on some hypothesis (linear elasticity) to infer stresses. See the report of Reviewer #2 for details.

We agree that the main text often refers to the SM for essential elements of our procedures and that sometimes this information should be present also in the main text. We now have taken several measurements to improve this situation: we added more explanations to the main text, we expanded the Materials and methods section, we have included the supplemental figures in the main document, and we have strongly rewritten and expanded the theory supplement, which is now also part of the main file (as appendix). We note, however, that the main text was purposefully written in a relatively compact form and that the figure captions contain some of the information that is not in the main text, because we wanted to avoid redundancies. Thus, also in the revision, we keep the relatively compact form of the main text, to focus on the essential results and not to clutter the manuscript with too many details. Except for some typos, that we did not mark, no changes have been performed to the figure captions. Title, abstract and introduction are also unchanged. All changes are clearly visible in the markup file.

We also have extended our discussion of MSM by adding new text at several locations, as suggested.

Please note that our work contains three types of figures: the main figures, the supplemental figures and the figures in the theory supplement (now appendix). In the theory supplement, we now have also added subsection to the two main section on continuum and contour model, in which we explain for which figures the two complementary models have been used.

Reviewer #2 (Recommendations for the authors):There are a few concerns. The linkage between the supplementary material and the main text can be improved (see point 1). To reproduce the theoretical panels, additional details are required (see point 1). Comparison of MSM stresses and theory (see point 2).

We thank reviewer #2 for this very positive assessment. We agree that integration between main text and SM was not perfect and we now improve this in the revised version. In the main text, we try to better explain which model has been used and why. We also have strongly expanded the theory supplement and in the theory supplement there are now explanations where in the main text the different models have been used. At the same time, we try to keep our main text compact and to avoid redundancies with the figure captions. We also clarify all other issues raised below. We are very grateful to reviewer #2 for going through our manuscript with such great care and for providing us with such a wealth of helpful comments.

1. The linkage between the supplementary information and the main text can be improved. The connection between some SM sections and the corresponding figures or claims in the main text is unclear in many cases. In many figures, it is unclear which was the theoretical description used, the underlying hypothesis made, or the values of parameters, which limits the reproducibility of some results.1.1. In Figure 2A, the authors provide a schematic of the contour plot. According to Equation 48 in SM, the total traction force F_s is a combination of the tension of the free edge, the tension (σ_x) of the vertical fiber, and a force F_ad (in the schematic is called F_a, is the same?). In the SM, F_ad is undefined.

We apologize for the confusion. Yes, these quantities are the same and we changed the notation in the SM according to the schematic in Figure 2A. Further, we have now defined the adhesion force F_a in the SM.

No details on how was σ_x estimated can be found.

We explained the procedure for estimating σ_xx in the description of Figure 2D. We now added a short description to the TFM and MSM subsection in the Materials and methods section.

Is there a contribution of σ_y in the force balance 48?

No, there is no direct contribution of σ_y to the force balance. Σ_y contributes indirectly through the inward-pull of the peripheral arc and hence influencing the line tension λ, following the theory set forward in Bischofs et al. PRL 2009. We added the explanation that σ_y does not directly contribute to the traction forces, but only indirectly through the inward pull of the arc and thus changing the direction in which the line tension λ pulls to Section 3.2 in the theory supplement.

In Section 2.1, the authors define σ_x and σ_y as anisotropic surface tension. Are these related to the stresses obtained from MSM?

The anisotropic surface tensions were obtained by applying the concepts of the contour model to shape measurements and TFM data. In this sense, they are not directly related to the MSM measurements, which focus on bulk and not boundaries. Yet we expected σ_x and σ_y to be correlated to the MSM measurements, and indeed, we then found a strong correlation between MSM measurements and anisotropic surface tensions in Figure 2D. We now added a corresponding explanation to the Results section of the main text, which we streamlined in general.

In Section 2.2.1, can the authors explain in more detail how was σ_x estimated from TFM data and explain how the center of the ellipse was set?

We now provide a more detailed description in the Materials and methods Section for σ_x and in the Theory Supplement for the ellipse center.

1.2. In Figure 3C, can the authors explain how were the two theoretical panels obtained and what parameter values used?

We used the 2D FEM simulations and obtained the traction maps by subtracting the baseline traction maps from the maps at peak strain energy. We have now added a section to the theory supplement and explain this in more detail.

In Section 1.4 of the SM, there is confusion between u, Y, and us and Ys. Can the authors clarify the physical definition of each of them? It is unclear to me that the tractions can be linearly related to the displacement of the substrate without approximations. In A. F. Mertz (2011), the authors write a similar expression by ignoring non-local elastic effects. Can the authors comment on this point? In Equations 21, the authors assume that the displacement of the cell and the substrate are equal but no justification for this approximation is provided. The derivation of the effective substrate height Equation 20 was not found. It is unclear why the substrate height depends on the cell layer size, can the authors comment on this point?

The underlying theory and approximations are based on the publication Mertz et al. PRL 2011 and Banerjee and Marchetti PRL 2012. Here the authors derive h_eff and Y_s. Indeed, our choice for Y_s is exactly the same as used by Mertz et al. PRL 2011.

It is correct that in the original manuscript we used the two symbols Y_s and Y for the same physical quantity. We introduced this notation to make clear that on the one hand Y can be expressed in terms of the cells Young<milestone-start />’<milestone-end />s modulus, Poisson<milestone-start />’<milestone-end />s ratio and force localization length and on the other hand is given by the substrate properties Es and h_s. We have now simplified the notation in the SM and omit the subscript <milestone-start />“<milestone-end />s” to avoid confusion.

1.3. In Figure 4, a scale of the active coupling f for panels B and E is missing. Can the authors add them?

We apologize for the omission. We now have added a color bar for the active coupling.

Can the authors provide a physical explanation for why the S_xx on the left half depends on f, and S_yy on the left half does not?

The force balance is very different in both directions: in x-direction the cells are in a tug of war with each other, but in y-direction, they essentially pull against the micropattern. Therefore, the active coupling interferes much more with the stress measured in the non-activated side in x-direction.

According to Equation (15), it seems that the Σ_opto is isotropic. It is unclear how do the author obtain a different active coupling for S_xx in panel C top and for S_yy in panel C bottom.

In Figure 5 we show that the active response of a doublet strongly depends on the polarization of the doublet. We therefore did not find it surprising that also within one cell, the active coupling can be different in one direction versus the other. Additionally, the quantification of the active coupling was only possible for the average cell and therefore we do not have any statistics on these measurements. The differences shown between S_xx and S_yy are fairly small and therefore might not be significant.

This comment applies also to Figure 5F and 6F. Can the authors provide more detail on how f was estimated?

We added a new part to the Materials and methods, explaining in detail how this quantification was performed.

Besides, I was not able to understand how were panels 4E and 4F obtained. Can the authors explain this in SM?

We added a paragraph to the Materials and methods, explaining in detail how this panels were plotted. Additionally, we would like to note, that all figure panels can be reproduced by running the code available on Github (https://github.com/ArturRuppel/ForceTransmissionInDoublets) with the data available on Dryad (https://datadryad.org/stash/dataset/doi:10.5061/dryad.sj3tx9683).

1.4. Similar comments apply to Supplementary figures, S4A, B and D, and S5C.

This is now also explained in the new subsection in the theory supplement.

Besides in S4, the authors use a Maxwell model instead of the Kelvin-Voigt that is used in the main text, however, no details on the SM could be found for the Maxwell model (values of the parameters, theoretical description, numerical solvers).

We apologize for this oversight and added the corresponding theoretical description to the theory supplement.

Several of the above concerns arise because the linkage between the SM and the main text can be improved. I would recommend making sections in the SM that focus in more detail on how each theoretical panel was obtained with enough detail so that these panels can be reproduced. Then these sections can be referenced in the main text or captions for the corresponding analyses. Note also that the order of SM sections does not correspond to the order of appearance in the main text.

We now structure the SM in two large sections, one for the continuum model and one for the contour model. We decided to keep this sequence because both models use FEM and it is easier to introduce this first for the continuum model and then for the contour model. In this way, the theory supplement is a self-sustained document in itself and of special interest to the readers with an interest in theory. In each of the two sections, we now explain which part of the main manuscript is concerned. Likewise in the main text, we now make better use of the theory supplement.

2. The authors use MSM data to compare with their theory. MSM is a method that infers stresses from traction force data by assuming that the material behaves as a linear elastic material with a fixed elastic modulus and Poisson ratio. In several parts of the main text, it should be clarified that MSM relays on some hypothesis (linear elasticity) to infer stresses (page 7 below Section "The presence of an intercellular…"). The MSM stress data are compared to theoretical predictions of a Kelvin-Voigt viscoelastic model with anisotropic active stresses. There are simple configurations where both models have different stress distributions for the same traction force distribution and the same boundary conditions. It is unclear to me that the two stresses can be expected to be comparable in the geometry of the experiments (2d square). Can the authors comment on this point?

It is correct that the underlying continuum description of MSM is linear elasticity. We now point this out and cite the relevant literature (Tambe PLOS One 2013 on "Monolayer Stress Microscopy: Limitations, Artifacts, and Accuracy of Recovered Intercellular Stresses”). We now also cite Bauer et al. PLOS Comp Biol 2021 for MSM, because we use their code, and the original paper on MSM, Tambe et al. Nature Materials 2011. Finally, we now cite Ng et al. *eLife* 2014, because they show that their MSM-like procedures make sense biologically (correlation between calculated stresses and protein localization).

In our own work, we initially used a Kelvin-Voigt model (following Anderson et al. BPJ 2023) to fit the time evolution of the experimentally measured strain energy. However, for our analysis it turned out that it is sufficient to only compare the baseline stress values and the stress values at maximal strain energy. Since the viscoelastic contribution is 0 for baseline and peak strain energy (extremal value -> d/dt = 0), we are essentially comparing linear elasticity with linear elasticity. The boundary conditions used for the MSM are the same as for the FEM (stress free boundaries). We added the sentences:

“Further, we note that at baseline and peak strain energy viscoelasticity does not contribute (extremal value -> d/dt = 0). Hence, the FEM data correspond to the case of pure linear elasticity and can be directly compared to the MSM measurements. The boundary conditions used for the MSM analysis are the same as for FEM model (stress free boundaries)” to Section 1.6 in the Theory Supplement.

[Editors’ note: what follows is the authors’ response to the second round of review.]Reviewer #3 (Recommendations for the authors):Because of our expertise and the request of the editors, we will focus our analysis only on the experimental part. This complements the reviews of the other reviewers that focused more on the computational modeling part.We think that this paper is compelling and provides significant advances in the field of epithelial biology. For this reason, we consider it appropriate for eLife and important for the scientific community in the field.However, we feel that some more controls and experiments are needed to fully support the claims of the article. We will be glad to recommend the article for publication if the authors can answer our concerns 1, 2, and 3 by experiments, and 4 and 5 by discussion.The optogenetic experiment (activation of the left cell and the observed contraction of the right cell) is central to this article, and the accidental activation of the right cells by light should be more carefully excluded. Indeed, various occurrences can lead to the accidental activation of the right cell, such as light scattering or protrusions. The demonstration that the authors provide in Figure 3 suppl 2, doesn't fully exclude accidental activation of the right cell.1. Protrusions. Cells could emit protrusions that go from the right cell underneath the left cell and thereby being activated when the left cell is stimulated. Especially in the context of Figure 5 where the degree of active coupling is not only correlated with structural/mechanical polarization, but also the length of the shared border between cells (larger area where protrusions might be accidentally activated in 2to1 patterns vs 1to2 patterns). This can be excluded by, for instance, using a genetically-encoded membrane marker expressed only in one of the two cells.2. Accidental light on the right cell. This could be easily confuted by having doublets with only the right cell expressing the optogenetic RhoA tool and illuminating the left cell that doesn't have the tool. In this case, no contraction should be observed in the right cell.

Let us address points 1 and 2 together, since they are closely related. Activation through stray light of the right cell is indeed a concern that we considered extensively in our work. The reviewer suggests either adding a membrane marker to the right cell or adding the optogenetic construct to only one of the cells. This is however not feasible, because doublets form most reproducibly when they form through cell division of a single cell (see Author response image 1; this is also the same strategy employed by Tseng et al. PNAS 2012 to obtain reproducible doublet geometries). Note that we never know for sure if a doublet started from one cell and divided or if they were one cell from the beginning, but that the difference is rather statistical. After ~6h of incubation and high seeding density (left), most doublets will have formed from two cells whereas after ~20h of incubation and low seeding density, most cells will have divided and thus most doublets will have started from a single cell.

**Author response image 1. sa2fig1:** 

More importantly, the asymmetry will be further exacerbated when using an opto and a non-opto cell, since the baseline contractility of the opto-cells are about twice as high than of non-opto cells (see Author response image 2, comparing opto-MDCK with MDCK Ecad-GFP, here called “WT”. See also Valon et al. Nature Comm. 2017 Figure 2g, where they found this to be true as well). Therefore, it is highly likely that the junction will be not central at all and much closer to the opto-cell. We conclude that this line of attack is not feasible.

In our work, we took a different route to exclude this possibility, which we partially presented in Figure 3 suppl2, but probably didn’t develop enough. We thought that the best way to show that there is not straylight activation of the right cell is to look at the recruitment of CRY2 in both cells as we activate only the left half. Our first experiment was to activate exactly the left half of the doublet with different intensities and then quantify the recruitment of CRY2 to the membrane (see Author response image 3). As you can see, recruitment saturates at around 0.9 mW/mm² and at that intensity, we already see some minor recruitment in the right cell. Note that it is, however only a fraction of the recruitment seen in the left cell.Next, to understand why this is happening, we measured the intensity profile of our DMD (digital micro mirror device) and saw, that right at the border of the activation region, the intensity drops to only 50% of the max. value and only strongly attenuates to 6% at 10 µm distance from the border (Figure 3 suppl2A). Therefore, we shifted the activation region by these 10 µm for all the experiments on doublets shown in the manuscript.

Now let’s compare these two situations: Activation of the left cell at 0.18 mW/mm² with the activation border right at the center. Here, the light intensity is about 50%, that is 0.09mW/mm², right at the junction and we see no CRY2 recruitment in the right cell. In the other situation, we activate the left cell with 0.9 mW/mm² with the activation border 10 µm shifted away from the center. Here, the right cell receives 6% of 0.9 mW/mm², that is 0.05 mW/mm², right at the junction. This is only half of the light compared to the first condition, where we already saw no recruitment. If there were a significant contribution of opto-RhoA activation through stray light, be it through cryptic lamellipodia, scattering or other effects, this should have been visible as a recruitment of CRY2 in the experiment in Figure 3 suppl2 for the lowest light intensity. Considering that even full activation of the whole doublet at this intensity of 0.05 mW/mm² did not yield a force response either, we consider the contribution of potential stray light to be negligible (Figure 3 suppl2B). We added the recruitment figure to the supplemental data and added more explanation of how we made sure that stray light activation is not an issue in the corresponding Results section.

3. If we assume a similar expression of optogenetic tool in each cell, the density of optogenetic receptors per area in doublets should be twice the density of singlets, leading to stronger activation of RhoA. Could this effect e.g. explain a larger increase of strain in doublets vs singlets? Or would results remain the same if light dose was halved in doublets?

We do not agree that it is clear that the relevant density should be exactly twice as high. We agree that singlet and doublet cover a similar area, so the projected density should be higher and ideally doubled, but in general, we cannot know how exactly the optogenetic molecules are distributed in regard to the force generating machinery. What we do know for sure is how much strain energy is generated on the substrate and our main result here is that doublets are much better in doing this than singlets for half-activation. We find our explanation about the flow gradient in singlets were plausible, but of course alternative hypotheses are possible. We have some data (which we did not include in the paper) that show that the higher density of optogenetic receptor (or optogenetic activator for that matter) cannot explain the difference between singlets and doublets. When we activate the full doublet and the full singlet, both respond very similarly. A real difference only comes up when only half of the singlet/doublet are activated (see Figure 3-figure supplement 2). We added the figure about global activation the supplemental data and added a short discussion about this point to the corresponding Results section.

4. While not clear what the molecular mechanism of transmission is (and out of the scope of this study), inhibition of e-cadherins binding (or similar) could be a way to disrupt force propagation into neighboring cells. If no force response is measured in the neighboring cell, this would also help the claim that the neighboring cell does not receive sufficient light to activate the optogenetic system. Also, such a result would demonstrate that cells require direct contact and no soluble factor is involved in the activation of the receiver. Have the authors done any attempt to perturb the signal propagation using drugs? If not, can the authors discuss this important aspect?

We certainly expect cadherins to play an important role in the force transmission we observe and certainly it would be very interesting to see what happens if you downregulate their expression. However, this aspect is not the focus of the current work and we left it for future studies. It is certainly not an easy experiment to do. The challenge is that full inhibition of cadherin expression would not lead to stable doublets, so any cadherin KO mutants are out of the question. Cadherin KD with e.g. siRNA might be an option, but these methods are difficult to titrate correctly and even then, it is unclear if one could obtain cohesive doublets, since the junction is under a lot of tension in our setup. Furthermore, discussions with colleagues who have tried to do cadherin KD but did not succeed satisfyingly have convinced us that we are not the right lab to attempt this experiment. One last alternative we tried was to use the construct published (An optochemical tool for light-induced dissociation of adherens junctions to control mechanical coupling between cells D Ollech et al., Nature communications 11 (1), 472, 2020), but the construct completely disrupted the junction and used the wavelengths that are necessary for the Rho Activation. In this way destroying the junction while inducing contraction is not possible with those systems. We now discuss this point shortly in the corresponding paragraph of the discussion.

5. On the formation of doublets: In the text, it is explained that: "most doublets have started as single cells and divided on the pattern to form a doublet."Was this difference controlled for? How did the rest of the doublets form? Were they seeded as doublets initially? Is there a factor (e.g. cell size) that biases which cells seed as doublets vs cells that seed alone?For the doublets formed after division: Is there any bias comparing cells after mitosis (in doublet) vs cells before mitosis (singlet), could cell cycle state explain e.g. the increased abundance of stress fibers in singlets? Thoughts about synchronizing cell cycle state?

When we do our experiments, we only start looking at our sample after about 16h to 20h of incubation time and then start looking for doublets. Therefore, we never know for sure if a specific doublet came from a single cell or if they were already two cells from the beginning. In the beginning of the project, we did large timelapse experiments with low magnification to better understand when and how we obtain symmetrical doublets with reproducible shape, and we found that we had most doublets after about 20h (see Author response image 3) and that most of the doublets we saw at this point had come from single cells. This assertion is therefore just statistical and it is impossible for us to make more precise statements about this. We added one of these timelapse video to the supplementary data.

**Author response image 3. sa2fig3:** (a) Composite brightfield image of opto-MDCK cells spreading on H-patterns. Fully spread doublets are marked with a white frame. (b) Graph shows number of fully spread doublets over time from one experiment on 20 kPa and one experiment on 40 kPa polyacrylamide gels.

Another point about cell seeding: The number of cells per pattern can be tuned on average, but there will always be a wide distribution of cell number per pattern. We know we have about 50.000 patterns on our sample, so this is how many cells we seeded every time. Since it is impossible to get perfectly homogeneous seeding, some patterns will have 0, most will have one and others will have two or more cells.

Concerning the cell cycle state, this could definitely have an impact on force transmission. E.g. it has been shown that cell contractility depends on cell cycle state (Variation in traction forces during cell cycle progression Benoit Vianay et al., 01 February 2018 https://doi.org/10.1111/boc.201800006). Synchronization could have maybe helped reduce the variance in our data, but MDCK cells are not easy to synchronize, so we decided to let statistics take care of any potential effect of the cell cycle.

Concerning stress fiber abundance in singlets vs. doublets, we have not quantified this. It seems to be the case that singlets have more, but we would rather speculate that the single cell is under higher tension than a single cell in the doublet, which leads to more stress fiber formation.

We clarified this point in the Cell Culture section in the Materials and methods part.

From the text, some questions remain regarding the method of selecting stimulation regions in optogenetic experiments.Were the regions selected automatically or manually?If the stimulation area was aligned using fluorescent Fibrinogen Alexa488, does this not activate the CRY2 system? If aligned on actin marker, how were stimulations kept of consistent area between repeats?How was stimulated, a single plane in confocal?

All optogenetic stimulations were performed with a DMD (digitial micromirror device) on an epifluorescence microscope. The cells were centered in the field of view using the channel with the microbeads, where the pattern is clearly visible, since the beads tend to accumulate around the pattern. The stimulation region was then drawn with reference to the pattern, with the border of the activation region being placed 10 µm away from the center of the pattern. In other words, we aligned the doublets with respect to the activation region, so we did not have to select an activation region for every doublet.

We added a small paragraph explaining this to the corresponding section in the Materials and methods part.